# PointMapPolicy: Structured Point Cloud Processing for Multi-Modal Imitation Learning

**Xiaogang Jia**[1*]   **Qian Wang**[1]   **Anrui Wang**[1]   **Han A. Wang**[2†]   **Balázs Gyenes**[1]
**Emiliyan Gospodinov**[1]   **Xinkai Jiang**[1]   **Ge Li**[1]   **Hongyi Zhou**[1]   **Weiran Liao**[1]
**Xi Huang**[1]   **Maximilian Beck**[3]   **Moritz Reuss**[1]   **Rudolf Lioutikov**[1]   **Gerhard Neumann**[1]

[1]Karlsruhe Institute of Technology   [2]Reality Labs, Meta   [3]Johannes Kepler University Linz

## Abstract

Robotic manipulation systems benefit from complementary sensing modalities, where each provides unique environmental information. Point clouds capture detailed geometric structure, while RGB images provide rich semantic context. Current point cloud methods struggle to capture fine-grained detail, especially for complex tasks, which RGB methods lack geometric awareness, which hinders their precision and generalization. We introduce PointMapPolicy, a novel approach that conditions diffusion policies on structured grids of points without downsampling. The resulting data type makes it easier to extract shape and spatial relationships from observations, and can be transformed between reference frames. Yet due to their structure in a regular grid, we enable the use of established computer vision techniques directly to 3D data. Using xLSTM as a backbone, our model efficiently fuses the point maps with RGB data for enhanced multi-modal perception. Through extensive experiments on the RoboCasa, CALVIN benchmarks and real robot evaluations, we demonstrate that our method achieves state-of-the-art performance across diverse manipulation tasks. The overview and demos are available on our project page.

## 1   Introduction

The advent of diffusion-based Imitation Learning (IL) has allowed robots to carry out complex, long-horizon tasks from raw image observations [1, 2]. RGB images are a common observation modality for diffusion policies due to their ubiquitousness and rich semantic information. However, policies conditioned on only RGB images lack 3D geometric information about the scene. This 3D information is crucial for learning generalizable policies that can act precisely in complex 3D scenes, especially when using multiple camera views [3–6]. An alternative modality is point clouds, unstructured sets of 3D points that preserve geometric shape, distances, and spatial relationships. In addition, points captured from multiple camera views can be transformed into a common reference frame and concatenated, yielding a natural and powerful way to fuse multiple cameras. Although numerous works use point clouds as an input modality [7–9], their irregular structure limits the network architectures that can be used with them. In contrast, RGB images are on a regular grid and can be processed using convolutional operators, but are susceptible to changes in perspective and lighting.

Current 3D processing approaches face fundamental limitations that create a critical gap between 3D geometric information and existing 2D vision architectures. Downsampling-based methods [7],

---

[*]Correspondence to jia266163@gmail.com

[†]This work is not related to Han A. Wang's position at Meta.

39th Conference on Neural Information Processing Systems (NeurIPS 2025).

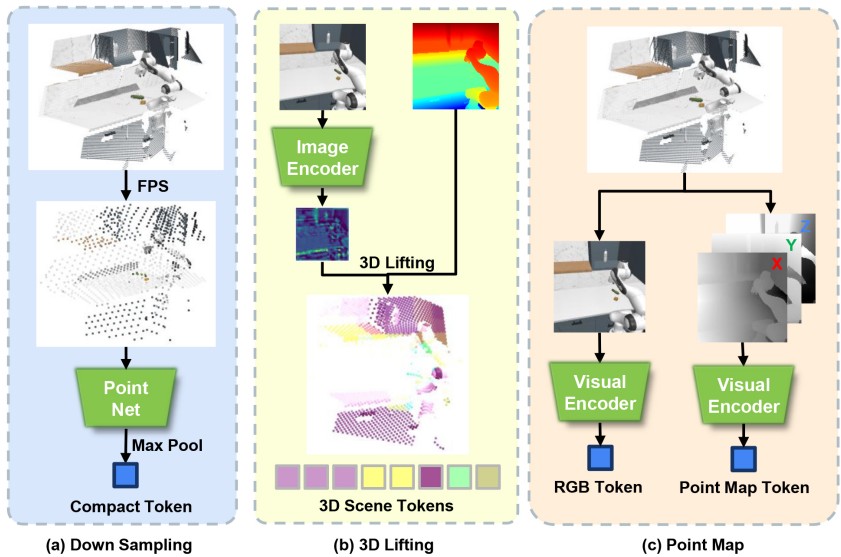

Figure 1: Different approaches for point cloud processing: **(a)** Downsampling-based methods use Furthest Point Sampling (FPS) to reduce dense point clouds to sparse representations, which PointNet then processes into compact tokens. Some variants employ FPS+KNN to generate structured point patches. **(b)** Feature-lifting approaches first extract 2D features from images, then project these features into 3D space, creating semantically rich 3D points. **(c)** Our point map method structures the point cloud as a 2D grid with the same dimensions as corresponding images, enabling direct application of efficient visual encoders to each modality independently.

as shown in Figure 1(a), suffer from an inherent information-fidelity tradeoff: they must dramatically reduce point density through techniques like Farthest Point Sampling (FPS)[10] to remain computationally tractable, inevitably discarding fine-grained geometric details essential for precise manipulation tasks. Feature-lifting approaches [11], as shown in Figure 1(b), face equally problematic limitations as they aggregate 2D features through depth averaging and 3D transformations, introducing information loss while struggling to maintain spatial structure during the lifting process. In this paper, we take inspiration from recent advances from the computer vision community in stereo reconstruction [12, 13] to propose using point maps, as shown in Figure 1(c). Point maps encode 3D information in a regular, 2D grid of Cartesian coordinates. This results in a structured data type that can be used with standard architectures such as ResNet [14], ViT [15], or ConvNeXt [16]. This obviates the need for steps like K-Nearest Neighors (KNN) and Farthest Point Sampling (FPS) [10], which are computationally expensive operations common to point cloud methods [17, 18, 6]. At the same time, because they are geometrically grounded, point maps from multiple views can be transformed into the same reference frame, increasing robustness to perturbations in camera perspective.

We integrate point maps into a standard diffusion-based imitation learning framework based on EDM [19] to demonstrate their effectiveness as a drop-in replacement for RGB images or point clouds. We validate the effectiveness of point map observations on two challenging benchmarks: RoboCasa [20] and CALVIN [21]. These benchmarks feature language-conditioned tasks and diverse scenes, and require spatial reasoning and long-term planning. Across both benchmarks, point map-based policies outperform baselines using RGB, depth maps, or point clouds, demonstrating superior learning efficiency and generalization. Our method is computationally efficient in training and inference, sometimes by an order of magnitude.

**Contributions:** Our contributions are the following: 1) we propose PointMapPolicy (PMP), a method for diffusion-based imitation learning on point maps, a powerful observation modality that has never been used in diffusion imitation learning; 2) we achieve state of the art results among policies trained from scratch on the CALVIN benchmark [21], and outperform other observation modalities on RoboCasa [20]; 3) we present systematic ablations of point cloud processing methods, vision backbones (e.g. ResNet [14], ViT [15], ConvNeXt [16]), and paradigms for fusing color and geometry information.

## 2   Related Work

**2D Visual Representations for Imitation Learning.** Recent imitation learning approaches [1, 22–25] rely predominantly on 2D visual representations such as RGB images or videos. Such representations are widely utilized due to their capacity to capture rich textural and semantic information, as well as their accessibility through low-cost cameras. However, 2D image modalities have inherent limitations: they contain 3D information only implicitly, are vulnerable to viewpoint and lighting changes and occlusions, and typically underperform in tasks requiring detailed spatial reasoning and geometric alignment [3–6].

**3D Visual Representations for Imitation Learning.** To overcome these limitations, a growing amount of research incorporates explicitly 3D representations such as depth maps, point clouds, or voxels. Voxel-based methods like C2F-ARM [26] and Perceiver-Actor [27] voxelize point clouds and use a 3D-convolutional network for action prediction, but require high voxel resolution for precision tasks, resulting in high memory consumption and slow training. DP3 [7] encodes sparse point clouds using FPS, followed by a lightweight MLP to produce a compact embedding vector of the observation. While efficient, this approach discards local geometric structure that can be critical for fine-grained tasks. In contrast, 3D Diffuser Actor [11] computes tokens by lifting 2D image features into 3D space by using averaged depth information and camera parameters, and applies FPS after the first cross-attention layer. FPV-Net [28] fuses RGB and point cloud modalities by injecting global and local image features into a point cloud encoder using adaptive normalization layers, but is still limited by the disadvantages of both modalities.

**Multi-View Representation.** Complementary work, such as Robot Vision Transformer (RVT) [29], avoids working directly with raw point clouds by proposing a novel multi-view representation. This approach re-renders the point cloud from a set of orthographic virtual cameras, deriving a 7-channel point map (RGBD + XYZ) from each view. RVT-2 [30] improves this approach for high-precision tasks by introducing a multi-stage inference pipeline: it first identifies a region-of-interest, truncates the observation to this area of interest, and then runs policy inference. However, neither of these methods use action diffusion, instead relying on key-frame based manipulation with a motion planner [27]. Furthermore, geometric and color information are fused naively at the channel level, whereas we investigate more sophisticated techniques for fusion.

**Diffusion-Policy Backbones.** Due to the non-Markovian nature of human demonstrations, where successful decision-making often depends on histories of past observations and actions, early work used RNN-based architectures [31], but struggled with vanishing gradients and limited scalability. This led to the adoption of Transformer-based architectures, which offer global attention and parallelism, enabling superior performance in tasks requiring long-horizon reasoning [32–34], becoming the standard backbone for many methods [7, 11, 28, 22]. However, Transformers are computationally intensive and scale quadratically with the sequence length, which limits the number of tokens that can be used to encode the observation.

To mitigate these challenges, recent works [35, 36] explore State Space Models (SSMs) like Mamba [37], achieving linear-time complexity and improved sample efficiency, particularly in low-data regimes. Additionally, recent recurrent architectures such as xLSTM [38] provide an appealing balance, maintaining the temporal modeling strengths of traditional RNNs while introducing architectural innovations that improve gradient flow and expressiveness. Despite being less expressive than self-attention, xLSTM significantly reduces compute and memory costs, making it well-suited for real-time or resource-constrained applications. X-IL [36] systematically compares different architectural parts, and finds that xLSTM performs competitively with Transformers in multi-modal imitation learning. Building on these insights, PointMapPolicy adopts xLSTM as its diffusion backbone, balancing temporal modeling capability with efficient training and inference.

## 3   Method

### 3.1   Problem Formulation

Imitation Learning aims to learn a policy from expert demonstrations. Given a dataset of expert trajectories $\mathcal{D}_\tau = \{\tau_i\}_{i=1}^N$, where each trajectory $\tau_i = ((\mathbf{s}_1, \mathbf{a}_1), (\mathbf{s}_2, \mathbf{a}_2), \ldots, (\mathbf{s}_K, \mathbf{a}_K))$. The objective is to learn a policy $\pi(\bar{\boldsymbol{a}}|\mathbf{s})$ that maps observations $\mathbf{s}$ to a sequence of actions $\bar{\boldsymbol{a}} = (\mathbf{a}_k, \mathbf{a}_{k+1}, \ldots, \mathbf{a}_{k+H})$. Predicting sequences of actions, i.e. action chunking, allows for more temporally consistent action

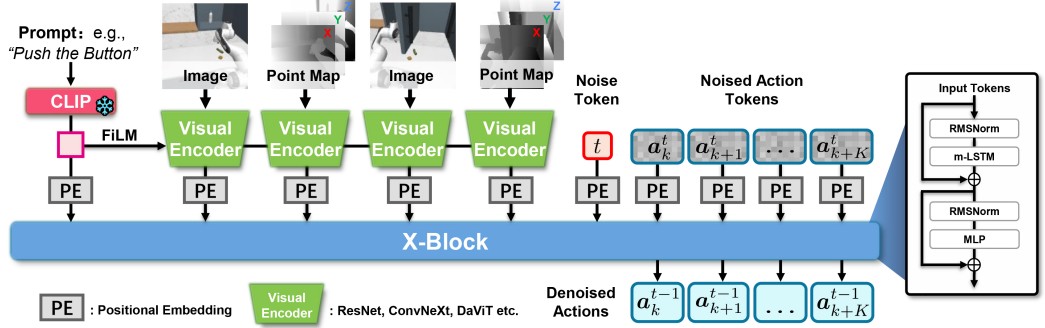

Figure 2: **Overview of PMP.** PMP integrates multiple modalities: language instructions encoded by a pretrained CLIP model, images processed by pretrained visual encoders, and point maps processed by visual encoders trained from scratch. Leveraging x-LSTM as its backbone architecture, PMP efficiently fuses these multi-modal inputs to generate denoised actions.

prediction [39]. Each observation $\mathbf{s}$ contains multi-view RGB-D images and language instruction for the current trajectory.

### 3.2 Score-based Diffusion Policy

Our approach employs the EDM framework for continuous-time action diffusion [19, 33] to generate actions. Diffusion models are generative models that learn to generate new samples through learning to reverse a Gaussian Perturbation process. In PointMapPolicy, we apply a score-based diffusion model to formulate the policy representation $\pi_\theta(\bar{\mathbf{a}}|\mathbf{s})$. This perturbation and its inverse process can be expressed through a Stochastic Differential Equation (SDE):

$$\mathrm{d}\bar{\boldsymbol{a}} = \left(\beta_t\sigma_t - \dot{\sigma}_t\right)\sigma_t\nabla_a \log p_t(\bar{\boldsymbol{a}}|\mathbf{s})dt + \sqrt{2\beta_t}\sigma_t d\omega_t, \tag{1}$$

where $\beta_t$ determines the noise injection rate, $d\omega_t$ represents infinitesimal Gaussian noise, and $p_t(\bar{\boldsymbol{a}}|\mathbf{s})$ denotes the score function of the diffusion process. It guides samples away from high-density regions during the forward process. To learn this score, we train a neural network $D_\theta$ via score matching [40]:

$$\mathcal{L}_{\mathrm{SM}} = \mathbb{E}_{\sigma,\bar{\boldsymbol{a}},\boldsymbol{\epsilon}}\left[\alpha(\sigma_t)|D_\theta(\bar{\boldsymbol{a}} + \boldsymbol{\epsilon}, \mathbf{s}, \sigma_t) - \bar{\boldsymbol{a}}|_2^2\right], \tag{2}$$

where $D_\theta(\bar{\boldsymbol{a}} + \boldsymbol{\epsilon}, \mathbf{s}, \sigma_t)$ represents our trainable neural architecture.

After training, we can generate new action sequences beginning with Gaussian noise by iteratively denoising the action sequence with a numerical Ordinary Differential Equation (ODE) solver. Our approach utilizes the DDIM-solver, a specialized numerical ODE-solver for diffusion models [41] that enables efficient action denoising in a minimal number of steps. Across all experiments, our method uses 4 denoising steps.

### 3.3 Observation Tokenization

We are given an observation $\mathbf{s}_k$ in step $k$ as well as a textual language instruction $z_{\mathrm{lang}}$. The language instruction is first tokenized using a pretrained CLIP text encoder [42] to generate language embeddings. For RGB inputs, we use Film-ResNet [43] with pretrained ImageNet weights to generate visual embeddings from the observation $\mathbf{s}_k$.

We define that a Point Map $X \in \mathbb{R}^{H \times W \times 3}$ is a dense 2D field of 3D points that establishes a one-to-one mapping between image pixels and 3D scene points. For an RGB image $I$ of resolution $H \times W$, the corresponding Point Map $X$ satisfies $I_{i,j} \leftrightarrow X_{i,j}$ for all pixel coordinates $(i, j) \in \{1 \ldots H\} \times \{1 \ldots W\}$, where each pixel intensity $I_{i,j}$ corresponds to a 3D point $X_{i,j} \in \mathbb{R}^3$ in world coordinates.

We convert each depth map $D \in \mathbb{R}^{H \times W}$ to a structured point map representation:

$$\mathbf{M}_t = \phi(D, K_{\mathrm{int}}^{-1}), \quad \mathbf{M}_t \in \mathbb{R}^{H \times W \times C} \tag{3}$$

where $K_{\text{int}}$ are the camera intrinsic parameters obtained through calibration and $\phi$ is a depth unprojection operation. The result is a multi-channel point map with the same spatial dimensions as the input depth map, where the channel dimension $C$ is typically 3. Points beyond a maximum depth and below a minimum depth are masked out. Point maps from all cameras are transformed into a common world reference frame using the extrinsic parameters of the camera $K_{\text{ext}}$.

### 3.4 PointMapPolicy

PointMapPolicy uses EDM-based action diffusion for decision making and conditions on the multimodal observation tokens generated from RGB and point map modalities. We present and explore multiple paradigms for fusing RGB and geometric data at various stages of processing. We also describe PMP-xyz, a variant with tokens from only the point map modality, for tasks that do not condition on color information.

**Fusion of image and point maps.** A key advantage of point maps is their ability to provide both geometric and visual embeddings for each camera view, enabling straightforward multimodal fusion. We investigate both early and late fusion approaches. For early fusion *PMP-6ch*, we concatenate point maps with RGB values, creating six-channel inputs (XYZ + RGB). For late fusion, we first tokenize image and point map modalities from each view with separate encoders.

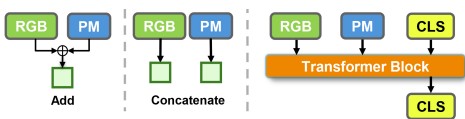

Figure 3: Fusion methods. From left to right: *Add*, *Cat*, and *Attn*.

Then we explore three methods to fuse encoded tokens, as illustrated in Figure 3: 1) *Add*, element-wise addition of tokens from both modalities, resulting in one token per view; 2) *Cat*, concatenation of tokens from all modalities and views; and 3) *Attn*, using a four-layer transformer module to process tokens using cross-attention to generate fused class tokens for each view. As shown in our ablation studies, we find *Cat* to slightly outperform other late fusion methods, so we choose this for PMP. An overview of PMP with *Cat* fusion is illustrated in Figure 2.

**Backbones.** Given the multi-modal tokens from Section 3.3, a learnable positional embedding is added to each token. PMP uses a decoder-only backbone from X-IL [36] with x-LSTM as the core computational unit. All tokens are concatenated as inputs to the X-Block, which is the diffusion score network $D_\theta$. While Transformers dominate most imitation learning policies, X-IL demonstrated that the recent recurrent architecture xLSTM excels in robot learning tasks. The core computational element within X-Block is the m-LSTM layer, which serves an analogous function to self-attention in Transformer architectures. The denoised action tokens produced by the X-Block are then used to guide the robot's behavior, resulting in a policy that effectively leverages both the geometric precision of point maps and the rich semantic understanding from RGB images.

## 4 Simulation Experiments

We conduct experiments on two simulation benchmarks RoboCasa [20] and CALVIN [21]. We aim to answer the following questions: **Q1)** How does PointMapPolicy compare to state-of-the-art 2D and 3D imitation learning policies? **Q2)** How do the fusion methods perform compared to other modalities? **Q3)** How does point map representation compare to other point cloud processing methods? **Q4)** Can current vision encoders effectively extract the geometric and semantic information from point maps required for robust decision-making?

**RoboCasa**: The RoboCasa benchmark [20] is a large-scale simulation framework designed to evaluate IL agents across a wide range of household manipulation tasks. Built on a physically realistic environment with rich visual rendering, RoboCasa supports task diversity, long-horizon behaviors, and fine-grained physical interactions, making it a compelling testbed for assessing both generalization and behavior diversity in policy learning. We use the RoboCasa benchmark to assess whether our proposed point map representation can enable effective learning and generalization across manipulation tasks of increasing complexity, object count, and behavioral variation.

**CALVIN**: The CALVIN benchmark [21] provides a large-scale framework for evaluating language-conditioned IL policies in visually rich, long-horizon manipulation tasks. The benchmark contains 34 distinct manipulation tasks such as button-pressing, drawer-opening, object-picking, and pushing.

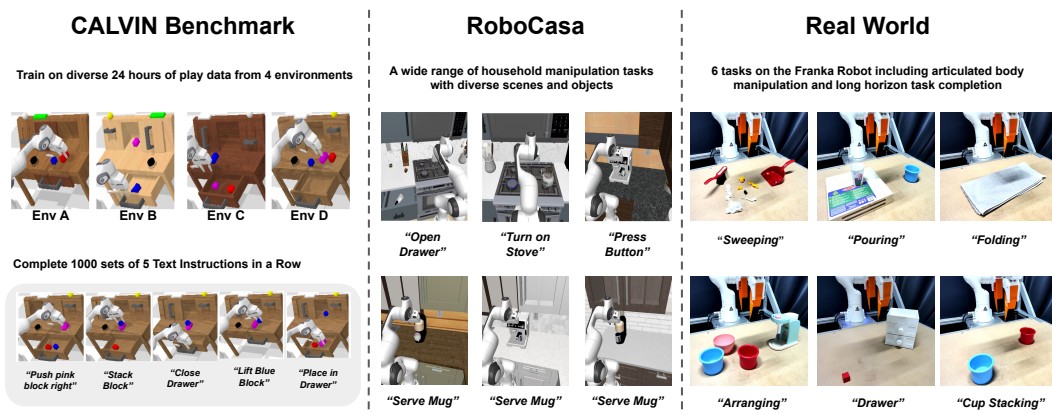

Figure 4: **Overview of Simulation and Real World Experiments used to test PointMapPolicy.** From left to right: CALVIN Benchmark [21], RoboCasa [20], and our Real World Setup.

| Category | Task | BC | GR00T-N1 | DP3 | 3DA | RGB | Depth | PMP-6ch | **PMP-xyz** | **PMP** |
|---|---|---|---|---|---|---|---|---|---|---|
| Pick and Place | PnPCounterToMicrowave | 2.0 | 0.0 | $4.0_{\pm1.6}$ | 0.0 | $10.0_{\pm4.3}$ | $3.3_{\pm0.9}$ | $9.3_{\pm0.9}$ | $\mathbf{13.3_{\pm3.4}}$ | $10.7_{\pm3.8}$ |
| | PnPCounterToSink | 2.0 | 1.0 | $0.7_{\pm0.9}$ | 0.0 | $5.3_{\pm1.9}$ | $4.7_{\pm0.9}$ | $\mathbf{8.7_{\pm0.9}}$ | $6.7_{\pm2.5}$ | $6.7_{\pm3.4}$ |
| | PnPMicrowaveToCounter | 2.0 | 0.0 | $4.0_{\pm2.8}$ | 0.0 | $10.7_{\pm3.8}$ | $8.0_{\pm1.6}$ | $12.0_{\pm1.6}$ | $\mathbf{16.0_{\pm1.6}}$ | $\mathbf{16.0_{\pm6.5}}$ |
| | PnPSinkToCounter | 8.0 | 5.9 | $1.3_{\pm0.9}$ | 0.0 | $14.7_{\pm0.9}$ | $3.3_{\pm0.9}$ | $9.3_{\pm4.7}$ | $8.0_{\pm1.6}$ | $\mathbf{16.7_{\pm5.0}}$ |
| Open/Close Drawers | OpenDrawer | 42.0 | 42.2 | $46.0_{\pm3.3}$ | 18.0 | $44.7_{\pm7.7}$ | $56.7_{\pm0.9}$ | $40.0_{\pm3.3}$ | $\mathbf{60.0_{\pm4.3}}$ | $56.0_{\pm8.2}$ |
| | CloseDrawer | 80.0 | **96.1** | $60.0_{\pm1.6}$ | 80.0 | $84.0_{\pm7.1}$ | $92.0_{\pm0.0}$ | $75.3_{\pm3.4}$ | $96.0_{\pm1.6}$ | $91.3_{\pm3.8}$ |
| Twisting Knobs | TurnOnStove | 32.0 | 25.5 | $24.7_{\pm4.1}$ | 18.0 | $18.7_{\pm1.9}$ | $22.7_{\pm0.9}$ | $35.3_{\pm2.5}$ | $\mathbf{43.3_{\pm5.7}}$ | $41.3_{\pm5.0}$ |
| | TurnOffStove | 4.0 | 15.7 | $7.3_{\pm1.9}$ | 8.0 | $13.3_{\pm0.9}$ | $14.0_{\pm1.6}$ | $16.7_{\pm2.5}$ | $\mathbf{20.0_{\pm3.3}}$ | $18.0_{\pm0.0}$ |
| Turning Levers | TurnOnSinkFaucet | 38.0 | 59.8 | $42.0_{\pm3.3}$ | 26.0 | $64.0_{\pm7.1}$ | $\mathbf{78.7_{\pm2.5}}$ | $64.7_{\pm4.1}$ | $76.7_{\pm4.1}$ | $66.7_{\pm8.1}$ |
| | TurnOffSinkFaucet | 50.0 | 67.7 | $42.0_{\pm4.9}$ | 44.0 | $63.3_{\pm7.7}$ | $76.0_{\pm5.9}$ | $73.3_{\pm9.6}$ | $\mathbf{82.0_{\pm1.6}}$ | $66.7_{\pm9.4}$ |
| | TurnSinkSpout | 54.0 | 42.2 | $58.7_{\pm6.8}$ | 28.0 | $50.0_{\pm4.9}$ | $76.0_{\pm4.9}$ | $69.3_{\pm3.8}$ | $76.0_{\pm1.6}$ | $48.7_{\pm7.4}$ |
| Pressing Buttons | CoffeePressButton | 48.0 | 56.9 | $14.7_{\pm0.9}$ | 8.0 | $70.7_{\pm13.7}$ | $84.0_{\pm4.3}$ | $76.7_{\pm6.6}$ | $82.7_{\pm5.0}$ | $\mathbf{92.0_{\pm3.3}}$ |
| | TurnOnMicrowave | 62.0 | **73.5** | $39.3_{\pm7.4}$ | 34.0 | $48.0_{\pm4.9}$ | $44.0_{\pm0.0}$ | $64.7_{\pm5.2}$ | $49.3_{\pm5.0}$ | $64.7_{\pm3.4}$ |
| | TurnOffMicrowave | 70.0 | 57.8 | $62.7_{\pm5.7}$ | 30.0 | $69.3_{\pm9.0}$ | $68.0_{\pm7.1}$ | $75.3_{\pm2.5}$ | $70.0_{\pm4.9}$ | $\mathbf{84.0_{\pm6.5}}$ |
| Insertion | CoffeeServeMug | 22.0 | 34.3 | $21.3_{\pm0.9}$ | 0.0 | $60.0_{\pm2.8}$ | $57.3_{\pm6.2}$ | $48.7_{\pm7.7}$ | $\mathbf{69.3_{\pm6.6}}$ | $49.3_{\pm3.4}$ |
| | CoffeeSetupMug | 0.0 | 2.0 | $4.0_{\pm2.8}$ | 2.0 | $16.0_{\pm2.8}$ | $15.3_{\pm0.9}$ | $10.7_{\pm0.9}$ | $16.7_{\pm3.8}$ | $\mathbf{26.7_{\pm3.4}}$ |
| **Average Success Rate** | | 32.25 | 36.28 | 27.04 | 18.50 | 40.16 | 44.00 | 43.12 | **49.12** | 47.22 |

Table 1: Success rate (%) for each task in RoboCasa [20]. The models were trained for 50 epochs with 50 human demonstrations per task and evaluated with 50 episodes for each task. The bold numbers highlight the best achieved success rate for that task among all the models.

Each rollout consists of a sequence of 5 language instructions, and the agent must complete one task before proceeding to the next. Policies are evaluated on 1,000 such instruction chains per seed, and success is measured by the average number of correctly completed tasks in each sequence.

**Experimental Setup**: For RoboCasa, each model was trained for 50 epochs using three random seeds, with performance measured at the $30th$, $40th$, and $50th$ checkpoints, selecting the best result. To ensure fair comparison, all models across different modalities use identical backbone parameters. For the CALVIN benchmark, models were trained for 25 epochs, with the best success rate reported from the $10th$, $15th$, $20th$, and $25th$ checkpoints.

**Baselines**: For RoboCasa, we benchmark against Behavioral Cloning (BC) [20], GR00T-N1 [44], 3D Diffusion Policy (DP3) [7], and 3D Diffuser Actor (3DA) [11]. Note that GR00T-N1 results use 100 demonstrations, while all other methods use 50 human demonstrations. To systematically evaluate the effectiveness of our representation, we further compare other against image-based baselines using only RGB data (RGB), and only depth data (Depth). We then compare PMP against multiple variants introduced in Section 3.4: *PMP-6ch* directly uses 6-channel point maps as inputs, and *PMP-xyz* only uses xyz coordinates as inputs. All five methods share the same architectures and parameters for fair comparison. Details can be found in Appendix 6.

For CALVIN, we primarily compare our approach against models without robot-specific pretraining, though we include all results for reference. DP3, 3DA, and CLOVER [45] are selected as representa-

| Train→Test | Method | PrT | Action Type | No. Instructions in a Row (1000 chains) | | | | | Avg. Len. |
|---|---|---|---|---|---|---|---|---|---|
| | | | | 1 | 2 | 3 | 4 | 5 | |
| | RoboFlamingo [48] | ✓ | Cont. | 82.4% | 61.9% | 46.6% | 33.1% | 23.5% | 2.47 |
| | SuSIE [49] | ✓ | Diffusion | 87.0% | 69.0% | 49.0% | 38.0% | 26.0% | 2.69 |
| | GR-1 [50] | ✓ | Cont. | 85.4% | 71.2% | 59.6% | 49.7% | 40.1% | 3.06 |
| | OpenVLA [23] | ✓ | Discrete | 91.3% | 77.8% | 62.0% | 52.1% | 43.5% | 3.27 |
| | RoboDual [51] | ✓ | Diffusion | 94.4% | 82.7% | 72.1% | 62.4% | 54.4% | 3.66 |
| | Seer [47] | ✓ | Cont. | 94.4% | 87.2% | 79.9% | 72.2% | 64.3% | 3.98 |
| | MoDE [24] | ✓ | Diffusion | 96.2% | 88.9% | 81.1% | 71.8% | 63.5% | 4.01 |
| | Seer-Large [47] | ✓ | Cont. | 96.3% | 91.6% | 86.1% | 80.3% | 74.0% | 4.28 |
| ABC→D | DP3 [7] | × | Diffusion | 28.7% | 2.7% | 0.0% | 0.0% | 0.0% | 0.31 |
| | MDT [46] | × | Diffusion | 63.1% | 42.9% | 24.7% | 15.1% | 9.1% | 1.55 |
| | 3DA [11] | × | Diffusion | 92.2% | 78.7% | 63.9% | 51.2% | 41.2% | 3.27 |
| | MoDE (scratch) [24] | × | Diffusion | 91.5% | 79.2% | 67.3% | 55.8% | 45.3% | 3.39 |
| | CLOVER [45] | × | Diffusion | 96.0% | 83.5% | 70.8% | 57.5% | 45.4% | 3.53 |
| | Seer (scratch) [47] | × | Cont. | 93.0% | 82.4% | 72.3% | 62.6% | 53.3% | 3.64 |
| | Seer-Large (scratch) [47] | × | Cont. | 92.7% | 84.6% | 76.1% | 68.9% | 60.3% | 3.83 |
| | RGB | × | Diffusion | 89.9% | 75.4% | 60.8% | 49.8% | 39.1% | 3.15 |
| | **PMP-xyz (ours)** | × | Diffusion | 73.0% | 51.9% | 37.0% | 24.5% | 16.1% | 2.03 |
| | **PMP (ours)** | × | Diffusion | **96.1%** | **88.6%** | **80.5%** | **72.3%** | **63.6%** | **4.01** |

Table 2: Evaluation results on the CALVIN benchmark under ABC→D. All results report the average rollout length averaged over 1000 instruction chains.

tive policies using RGB-D inputs. MDT [46], MoDE [24], and Seer [47] are selected as RGB-based policies. Further details of these baselines can be found in Appendix A.2.

## 4.1 Main Results

**RoboCasa.** We present the main results in Table 1. PMP-xyz demonstrates significant advantages over prior 3D baselines DP3 and 3DA, achieving an average success rate of 49.12%—nearly 20% higher. It also outperforms 2D baselines BC and GR00T-N1 by approximately 13%. The above results address **Q1**. Our cross-modality evaluation using consistent architectures reveals that incorporating 3D information consistently improves performance in RoboCasa. Specifically, PMP-xyz shows a 6% improvement over the Depth-only model, highlighting the value of structured point maps. While PMP (47.22%) outperforms PMP-6ch, demonstrating the benefits of late fusion, it still falls 2% short of PMP-xyz. This pattern suggests that most RoboCasa tasks favor geometric information, likely due to the diversity of objects and scenes.

**CALVIN.** On the CALVIN benchmark, PMP achieves a score of 4.01, outperforming all other models trained from scratch and many models that leverage pretrained data, as shown in Table 2. Our method even outperforms Seer-Large (scratch) which scores 3.83 despite using 24 Transformer layers compared to our smaller model using only 10 x-LSTM blocks. This answers **Q1** in the affirmative.

PMP-xyz performs poorly with an average rollout length of only 2.03, while the RGB-only model achieves a respectable score of 3.15. This performance disparity stems from CALVIN's heavy reliance on color information for task execution, with many instructions explicitly referencing colors (e.g., "red block", "pink block"). This finding highlights a key limitation of purely geometric representations in color-dependent scenarios.

These contrasting results between RoboCasa and CALVIN benchmarks underscore the complementary nature of geometric and visual information. While PMP-xyz excels in geometry-heavy tasks (RoboCasa), it struggles with color-dependent tasks (CALVIN). This demonstrates that multimodal fusion approaches like PMP provide the most robust and versatile performance across diverse task domains by adaptively leveraging the most relevant modality for each scenario, addressing **Q2**.

## 4.2 Point Cloud Encoding

While our main results demonstrate the effectiveness of PMP compared to other methods, these comparisons involve different policy backbone architectures. To isolate the contribution of our point cloud encoding approach, we conduct a controlled ablation study where we fix the policy backbone (X-Block) and systematically vary only the point cloud encoder.

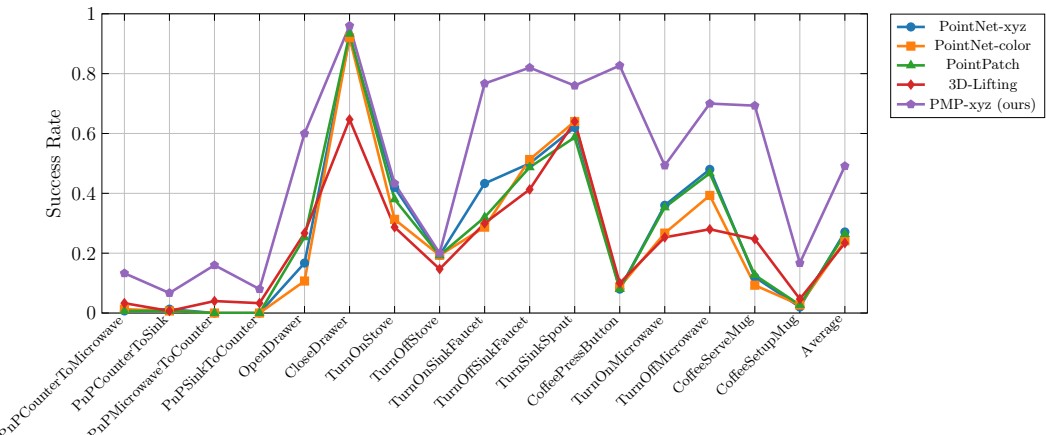

Figure 5: Ablation study comparing point cloud encoders with fixed X-Block policy backbone. Our PMP-xyz method substantially outperforms baseline encoders across all manipulation tasks, demonstrating that our improvements arise from the point cloud encoding approach.

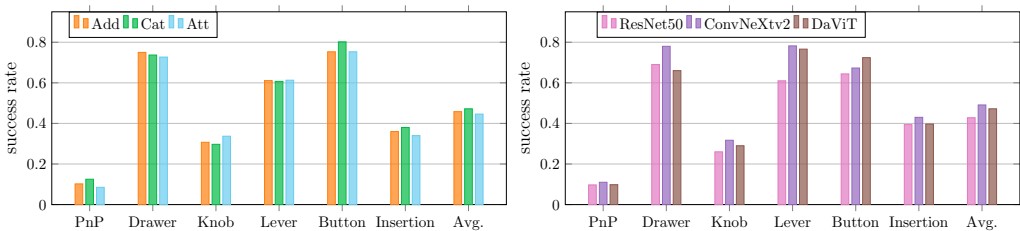

Figure 6: *Left*: Performance comparison of various fusion methods between point maps and images. *Right*: Performance comparison of different visual encoders for point map processing.

We conducted controlled experiments on RoboCasa using identical xLSTM backbones with different point cloud processing encoders: 1) PointNet-xyz: Following DP3 [7], we gather point clouds from 3 camera views and use Furthest Point Sampling (FPS) to downsample to 1024 points, then apply MLP with maxpooling to create a compact 3D token. 2) PointNet-color: Same process as PointNet-xyz but using colored points with XYZRGB information. 3) PointPatch: We use FPS to sample 256 center points, apply k-Nearest Neighbors to create 256 point patches with 32 points each, tokenize each patch using MLP with maxpooling, then process the resulting tokens with a transformer to generate compact 3D representations. 4) 3D-Lifting: We extract CLIP features (frozen) from each camera view and lift the 2D features into 3D space, then use a transformer to process the lifted tokens. The 3D tokens are then passed to the diffusion policy with an identical X-Block backbone.

Figure 5 presents the success rates over 16 RoboCasa tasks. Our PMP-xyz achieves an average success rate of 49.12%, substantially outperforming all baselines. The consistent improvements demonstrate that the point maps approach effectively captures the spatial understanding necessary for robotic manipulation, independent of the downstream policy architecture, addressing **Q3**.

### 4.3 Ablation Study

We additionally conduct three ablations across the 6 categories of RoboCasa:

**Fusion of Images and Point Maps.** One key advantage of point maps over traditional point cloud processing methods is their structural similarity to RGB images from corresponding views. This alignment enables direct fusion of visual representations with point cloud data on a per-view basis. We evaluate three fusion strategies—*Add*, *Cat*, and *Attn*—which are described in Figure 3. The comparative performance of these fusion strategies is presented in Figure 6. Although the performance differences are modest, *Cat* consistently emerges as the most effective fusion approach.

**Vision Encoders for Point Maps.** To assess how well existing visual architectures process Point Map representations, we conduct a comparative analysis of three prominent visual encoders: FiLM-ResNet50, ConvNeXt-v2, and DaViT. The results in Figure 6 demonstrate that while all encoders can effectively process point maps, ConvNeXtv2 consistently outperforms the others across all RoboCasa tasks, addressing **Q4**.

**Understanding Model Attention Patterns.** To uncover where the model attends during action prediction, we apply Grad-CAM++ [52] to highlight the regions most influential for action decisions across different modalities. For detailed visualizations, see Appendix D.

## 4.4 Computation Resources and Inference Time

For the CALVIN experiments, PMP employs Film-ResNet50 as encoders for both images and point maps, with 8 x-Blocks as backbones (512 latent dimensions), totaling 147M trainable parameters. Training utilizes 4 Nvidia RTX 6000 Ada GPUs with 128 samples per GPU (512 total batch size). Each epoch completes in approximately 13 minutes, allowing full training (25 epochs) in under 6 hours, excluding evaluation time. More details can be found in Appendix E.

Regarding computational efficiency, we conducted inference latency benchmarks for our models using ConvNeXt-v2 encoders on a single Nnidia RTX 5080 GPU (batch size 1). Across 1000 prediction cycles, PMP-xyz demonstrates remarkable efficiency with an average inference time of 2.9 ms, while PMP requires only 3.9 ms, maintaining real-time performance.

## 5 Real-World Experiments

We evaluate PMP on six challenging real-world robot manipulation tasks: Arranging, Folding, Cup-Stacking, Drawer, Pouring, and Sweeping. An overview of our robot setup is shown in Figure 7. The robot's perception system consists of two RGB-D cameras mounted on the left and right sides of the workspace.

### 5.1 Real-World Benchmark

**Real-world Setup.** We evaluate our policies on a 7-DOF Franka Panda robot in six challenging tasks. Visual information is captured by two Orbbec Femto Bolt cameras, positioned to provide left and right views. These sensors provide both RGB and depth images, which are used to generate calibrated 3D point clouds. All RGB and depth images are resized to 180×320 resolution. The robot operates in an 8-dimensional action space, including joint positions and gripper state.

**Datasets.** For collecting demonstrations, we use a teleoperation system consisting of a leader robot and a follower robot. For each task, we collect varying numbers of language-conditioned trajectories as detailed in Table 3. To ensure robust evaluation, we randomly initialize the object and goal states, introducing significant variation in the objects used. For instance, in the sweep task, the broom can appear in 10 different areas, and the garbage in 4 different areas. In addition, the number, positions, and even categories of trash items are varied in the collection and evaluation.

### 5.2 Baselines and Metrics

**Baseline.** To evaluate the effectiveness of the point-map representation, we benchmark our methods against RGB-only policy, sharing with the same backbone. Each method is evaluated over 20 trials per task at training checkpoints 70,000, 80,000, and 90,000, using randomized initial object states to ensure robustness. We report results from the best-performing checkpoint for each method.

**Metrics.** Given the long-horizon nature and complexity of the tasks, we introduce a structured scoring metric to enable fair and detailed comparisons. Each task is decomposed into multiple stages, with each successfully completed stage contributing 1 point to the overall score. The final task score is the sum of the completed intermediate stages, providing a more granular measure of progress and policy effectiveness. The details of our scoring metrics can be found in Table 5.

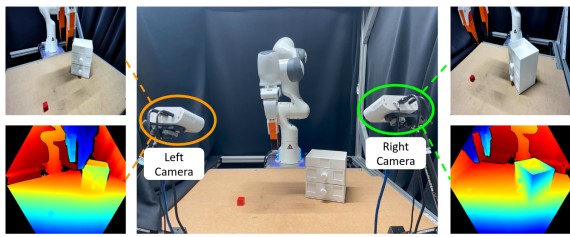

Table 3: The table shows average completed stages. The Max. indicates the total number of stages per task.

| Tasks | Demos Per Task | Methods with Scores | | | |
|---|---|---|---|---|---|
| | | RGB | PMP-xyz | PMP | Max |
| Arranging | 80 | 2.05 | 2.10 | **2.25** | 3 |
| Folding | 45 | 2.1 | 0.80 | **2.50** | 3 |
| Cup-Stacking | 75 | 1.40 | 0.45 | **2.10** | 3 |
| Drawer | 120 | 2.00 | 2.15 | **2.40** | 4 |
| Pouring | 80 | 1.55 | 1.60 | **1.80** | 4 |
| Sweeping | 90 | 1.80 | 0.80 | **2.15** | 4 |

Figure 7: Real world experiments consisting of six tasks. The left figure shows our setup and the Drawer task. The Table shows the average completed stages out of 20 evaluations.

## 5.3 Real-World Main Results

We evaluate all the methods with 20 rollouts per task. As can be seen in Table 3, our proposed PMP policy consistently outperforms all baselines across all evaluated real-world tasks. Compared to the RGB-only policy, PMP achieves at least a 0.2-point improvement in accumulated scores, demonstrating the effectiveness of fusing both point-map and RGB modalities. Notably, on the Folding task, PMP increases the score from 2.1 to 2.5 using only 45 demonstrations, showcasing strong sample efficiency.

Interestingly, the PMP-xyz also outperforms the RGB-only baseline on several tasks, underscoring the value of spatial structure in guiding action prediction. However, its performance drops significantly in tasks involving deformable objects, such as Folding and Sweeping, where object geometry would change over actions. In these scenarios, the lack of appearance cues leads to coarse and less reliable action predictions. This is especially evident in Cup-Stacking, a task that explicitly requires reasoning about object color, further highlighting the importance of RGB input. Overall, these results validate the effectiveness and generalizability of PMP in handling diverse and challenging manipulation tasks in the real world.

## 6 Limitation and Future Work

Our current approach has two main limitations. First, simply concatenating the point map and RGB tokens may not optimally leverage the complementary information in each modality. More sophisticated fusion mechanisms could potentially extract richer cross-modal relationships and further improve performance. Second, our point map visual encoders are trained entirely from scratch, which constrains their performance compared to the RGB modality that benefits from ImageNet pretraining. For future work, developing pretraining objectives specifically designed for point map encoders represents a promising direction. Just as vision models benefit substantially from pretraining on large image datasets, establishing similar paradigms for point map representations could dramatically improve performance, enabling more robust geometric feature learning before fine-tuning on specific robotic tasks.

## 7 Conclusion

We present PointMapPolicy (PMP), a novel diffusion-based imitation learning framework that effectively integrates 3D geometric reasoning with standard vision techniques. By projecting depth pixels into a multi-channel image of XYZ coordinates, PMP leverages existing visual encoders, while an efficient xLSTM-based diffusion network denoises action tokens to generate precise control sequences. Empirical results on RoboCasa and CALVIN demonstrate that PMP not only achieves state-of-the-art performance but also offers significantly faster training and inference. Comprehensive ablations on observation modalities and fusion strategies further highlight the clear advantages of structured point-map representations. Looking forward, we plan to explore large-scale pretraining of point-map models to extend generalization across diverse robotic tasks.

# 8 Acknowledgments

This work was supported by the European Research Council (ERC) under the European Union's Horizon Europe programme through the project SMARTI³ (Grant Agreement No. 101171393). The authors also acknowledge funding from the German Research Foundation (DFG) within the framework of Collaborative Research Centre SFB 1574 "Circular Factory for the Perpetual Product" (Project No. 471687386). This work was also supported by funding from the pilot program Core Informatics of the Helmholtz Association (HGF). NS and GN were supported by the Carl Zeiss Foundation under the project JuBot (Jung Bleiben mit Robotern). Xiaogang Jia and Xinkai Jiang acknowledge the support from the China Scholarship Council (CSC). The authors also acknowledge support by the state of Baden-Württemberg through the HoreKa supercomputer funded by the Ministry of Science, Research and the Arts Baden-Württemberg, and by the German Federal Ministry of Education and Research.

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

# A Simulation Experiment Details

## A.1 RoboCasa Benchmark

RoboCasa [20] is a large-scale simulation framework developed to train generalist robots in realistic and diverse home environments, with a particular focus on kitchen scenarios. The benchmark comprises 100 tasks, including 25 atomic tasks with 50 human demonstrations and 75 composite tasks with auto-generated demonstrations. These tasks are centered around eight fundamental robotic skills relevant to real-world home environments: (1) pick-and-place, (2) opening and closing doors, (3) opening and closing drawers, (4) twisting knobs, (5) turning levers, (6) pressing buttons, (7) insertion, and (8) navigation.

To comprehensively evaluate our method, we selected five tasks from the atomic tasks described in Table 4, each representing a distinct skill.

| Task Name | Description |
|---|---|
| **Pick-and-Place Tasks** | |
| *PickPlace_Counter_To_Microwave* | Pick an object from the counter and place it inside the microwave (door is open). |
| *PickPlace_Counter_To_Sink* | Pick an object from the counter and place it in the sink. |
| *PickPlace_Microwave_To_Counter* | Pick an object from the microwave and place it on the counter (door is open). |
| *PickPlace_Sink_To_Counter* | Pick an object from the sink and place it on the counter next to the sink. |
| **Drawer Tasks** | |
| *Open_Drawer* | Open a drawer. |
| *Close_Drawer* | Close a drawer. |
| **Stove Tasks** | |
| *Stove_On* | Turn on a specific stove burner by twisting its knob. |
| *Stove_Off* | Turn off a specific stove burner by twisting its knob. |
| **Sink Tasks** | |
| *SinkFaucet_On* | Turn on the sink faucet to start water flow. |
| *SinkFaucet_Off* | Turn off the sink faucet to stop water flow. |
| *Turn_Sink_Spout* | Rotate the sink spout. |
| **Coffee Machine Tasks** | |
| *Coffee_Press_Button* | Press the button to pour coffee into the mug. |
| *Coffee_Setup_Mug* | Place the mug into the coffee machine's mug holder. |
| *Coffee_Serve_Mug* | Remove the mug from the holder and place it on the counter. |
| **Microwave Tasks** | |
| *Microwave_On* | Start the microwave by pressing the start button. |
| *Microwave_Off* | Stop the microwave by pressing the stop button. |

Table 4: RoboCasa task set.

## A.2 CALVIN Benchmark

**Benchmark Setup.** The CALVIN benchmark [21] is a long-horizon manipulation benchmark featuring four visually distinct tabletop environments (A–D), each containing a common set of objects and 34 manipulation tasks. Agents are given natural language instructions describing sequences of up to 5 tasks to be executed in order. The primary evaluation involves completing 1000 such instruction chains in environment D. Agents are scored by the number of tasks successfully completed per chain, with a maximum rollout length of 5.

**Evaluation Protocol.** We evaluate PointMapPolicy on one standard CALVIN settings: **ABC→D**, where the policy is trained on environments A, B, and C, and evaluated zero-shot on D. Only 1% of the play data is paired with language, requiring models to learn from primarily unlabeled data. The ABC→D setup tests visual and environmental generalization, while D→D emphasizes efficiency in low-resource, language-scarce settings.

**Baselines.** We compare against a broad set of state-of-the-art language-conditioned policies spanning imitation, diffusion, and foundation-model-based architectures:

- **RoboFlamingo** [48]: based on OpenFlamingo, this model integrates a frozen VLM with a lightweight policy head. It is pretrained on large-scale vision-language data and finetuned on CALVIN using supervised behavior cloning.

| Drawer | Stack | Fold | Score |
|--------|-------|------|-------|
| Open the upper/lower drawer | Pick the correct cup | Pick the towel | 1 |
| Pick the object and place it into drawer | Stack failed | Fold the towel | 2 |
| Close the Drawer | Stack the cups | Fold the towel perfectly | 3 |
| – | – | – | 4 |

| Pour | Sweep | Arrange | Score |
|------|-------|---------|-------|
| Pick the cup | Pick the broom | Open the mixer machine | 1 |
| Pour the contents | Sweep 20% of garbage | Pick the container and place on pad | 2 |
| Pour all contents into container | Sweep 50% of garbage | Close the mixer machine | 3 |
| Put the cup back | Sweep all garbage | – | 4 |

Table 5: Task score metric details of real robot and evaluation standards.

- **SuSIE** [49]: a scalable instruction-following diffusion policy. It is pre-trained on curated robot demonstrations and uses an instruction-conditioned denoising process with significant offline finetuning.
- **GR-1** [50]: a powerful decoder-only transformer trained on large-scale synthetic video data. The model is capable of generating long sequences of actions and is finetuned on CALVIN for grounding.
- **CLOVER** [45]: a video diffusion planner that predicts intermediate visual goals via video generation and closes the loop using low-level policy feedback. It does not require internet-scale pretraining and achieves strong multi-step rollout success.
- **MoDE** [24]: Mixture-of-Diffusion-Experts model with sparse routing. It supports both small (non-pretrained) and large (pretrained) variants. The pretrained variant achieves top performance while maintaining low inference cost.
- **Seer / Seer-Large** [47]: large-scale transformer models pretrained on 1000+ hours of robot play data. Seer incorporates language, vision, and action streams into a unified transformer and achieves strong generalization, particularly when scaled up.

# B    Real World Experiment Details

We conducted six real-world experiments on a Franka Panda Robot: Drawer, Stack, Pour, Sweep, Fold, and Arrange.

## B.1    Task Metric

Given the complexity and long-horizon nature of the tasks, we decompose each task into several discrete stages. The final score is computed as the total number of successfully completed stages. Details of the scoring metric design are provided in Table 5.

## B.2    Task Description

**Drawer**: In the Drawer task, there is a cabinet with two drawers and two different objects, a cube and a cylinder. The robot must follow a language-specified instruction to open the designated drawer, pick up the target object, place it inside the drawer, and then close the drawer. The key challenges involve handling the random initialization of both the cabinet's position and the objects' locations.

**Stack**: In the Stack task, four cups of different colors and sizes are provided. The robot must stack the cups in a specific order based on their colors. The main challenges lie in accurately recalling the stacking sequence and executing precise placement, as the cups are closely sized and must fit together properly.

**Pour**: In the Pour task, three distinct cups and three different containers are placed in randomized initial positions. The robot must generalize to novel object configurations while maintaining the

precision necessary to pour the contents from the cups into the containers without spilling. The primary challenge lies in adapting to varying spatial arrangements while executing controlled and accurate pouring motions.

**Sweep**: Unlike standard Pick-and-Place tasks, this task requires the robot to acquire a novel sweeping skill. In the Sweep task, the positions of the broom, dustpan, and trash vary across trials, and even the number of trash items changes. The key challenge is manipulating deformable trash materials that differ from those encountered during training, requiring the policy to exhibit strong generalization and adaptability.

**Fold**: The Fold task requires precise manipulation skills. The goal is to neatly fold a towel that is randomly oriented at the start of each trial. The primary challenge lies in accurately handling the soft, deformable material to achieve a clean and consistent fold despite varying initial conditions.

**Arrange**: In the Arrange task, the setup includes a mixing machine and a container. The robot must follow a specific sequence: first, open the mixing machine; next, place the container on the designated pad; and finally, close the machine. This task primarily emphasizes long-horizon planning, requiring the robot to execute a multi-step procedure in the correct order.

## C  Hyper Parameters

| Hyperparameter | CALVIN ABC | RoboCasa | Real World |
|---|---|---|---|
| Number of x-Blocks | 10 | 8 | 6 |
| Attention Heads | 8 | 8 | 8 |
| Action Chunk Size | 10 | 10 | 10 |
| History Length | 1 | 1 | 1 |
| Embedding Dimension | 2048 | 768 | 2048 |
| Image Encoder | FiLM-ResNet50 | ConvNextV2 | FiLM-ResNet50 |
| Goal Lang Encoder | CLIP ViT-B/32 | CLIP ViT-B/32 | CLIP ViT-B/32 |
| Attention Dropout | 0.3 | 0.3 | 0.3 |
| Residual Dropout | 0.1 | 0.1 | 0.1 |
| MLP Dropout | 0.1 | 0.1 | 0.1 |
| Optimizer | AdamW | AdamW | AdamW |
| Betas | [0.9, 0.95] | [0.9, 0.95] | [0.9, 0.95] |
| Learning Rate | 1e-4 | 1e-4 | 1e-4 |
| Transformer Weight Decay | 0.05 | 0.05 | 0.05 |
| Other weight decay | 0.05 | 0.05 | 0.05 |
| Batch Size | 128 | 128 | 128 |
| Train Steps in Thousands | 25 | 15 | 30 |
| $\sigma_{max}$ | 80 | 80 | 80 |
| $\sigma_{min}$ | 0.001 | 0.001 | 0.001 |
| $\sigma_t$ | 0.5 | 0.5 | 0.5 |
| EMA | True | True | True |
| Time steps | Exponential | Exponential | Exponential |
| Sampler | DDIM | DDIM | DDIM |
| Trainable Parameters (Millions) | 147 | 111 | 96 |

Table 6: Summary of all the Hyperparameters for our experiments.

We export all the hyper parameters across three benchmarks for reproduction.

## D  Activation Map Analysis

To gain qualitative insights into what regions the visual encoders attend to during action inference, we visualize activation maps using Grad-CAM++ [52]. Unlike classification tasks, our diffusion-based policy does not predict discrete categories, therefore, we apply Grad-CAM++ using the diffusion loss as the target signal, following the approach of highlighting input regions that most influence the denoised trajectory prediction. We generate the heatmaps using the Grad-CAM++ implementation[3], and compute activations for each camera view across three RoboCasa tasks: OpenDrawer, Turn On Sink Faucet, and Coffee Serve Mug. In all figures, we use a ConvNeXtv2 encoder and extract Grad-CAM++ heatmaps from the final convolutional block before normalization. Each visualization consists of six images arranged in two rows. The top row shows the raw visual input (RGB or XYZ visualized in color), and the bottom row displays the corresponding Grad-CAM++ heatmaps for each

---
[3] https://github.com/jacobgil/pytorch-grad-cam

of the three camera views: static left, static right, and wrist-mounted. These maps highlight spatial regions with the greatest impact on predicted actions.

Overall, the attention patterns are consistent with task-relevant visual cues. For example, activations commonly focus on the robot gripper, the manipulated object, or the goal location, depending on the modality and perspective.

Figures 8, 9 and 10 show results for the RoboCasa tasks Coffee Serve Mug, Open Drawer and Turn On Sink Faucet, respectively.

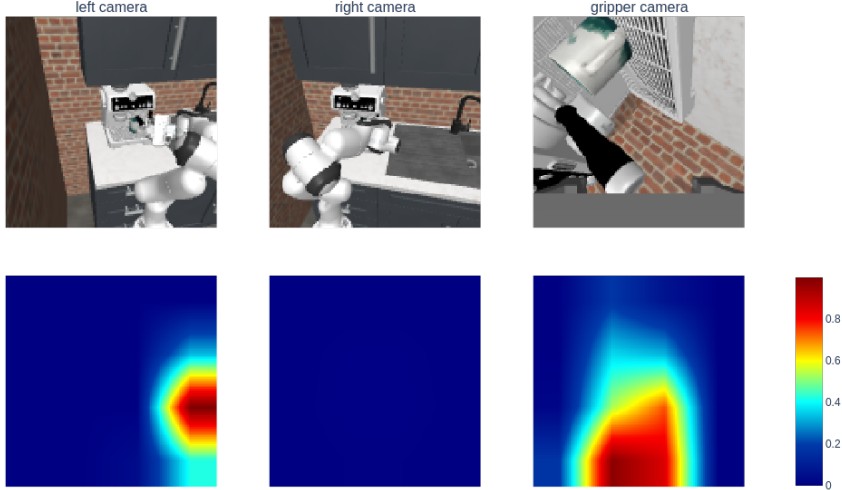

(a) RGB-only ConvNeXtv2 encoder. Top: raw 128x128 RGB input frames provided to the agent. Bottom: Grad-CAM++ heatmaps from the final convolutional layer.

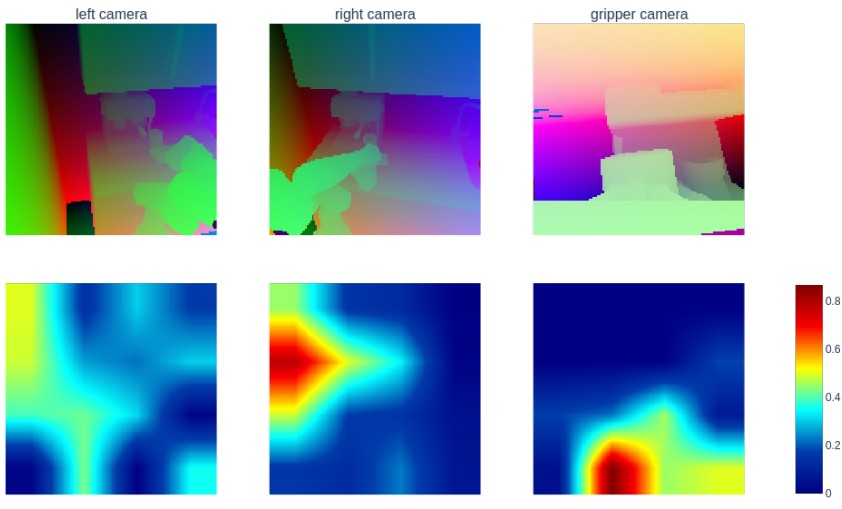

(b) PMP-xyz ConvNeXtv2 encoder. Top: 128x128 XYZ input visualized as RGB. Bottom: Grad-CAM++ heatmaps from the final convolutional layer.

Figure 8: Raw RGB, XYZ input frames and Grad-CAM++ activations on the Coffee Serve Mug RoboCasa task for RGB-only and Point-map-only ConvNeXtv2 visual encoders.

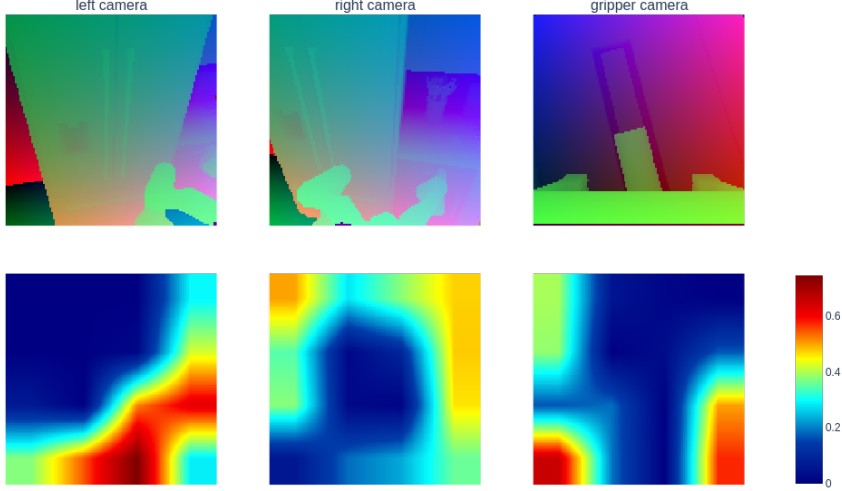

(a) RGB-only ConvNeXtv2 encoder. Top: raw 128x128 RGB input frames provided to the agent. Bottom: Grad-CAM++ heatmaps from the final convolutional layer.

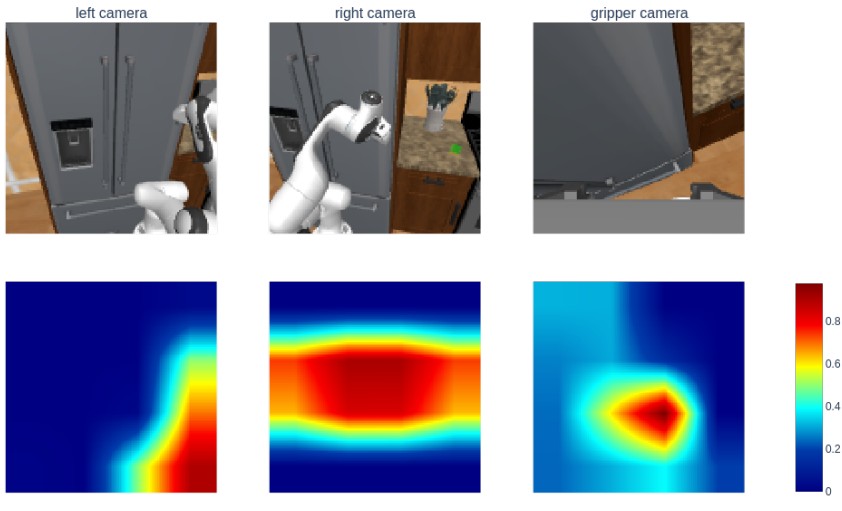

(b) PMP-xyz ConvNeXtv2 encoder. Top: XYZ input visualized as RGB. Bottom: Grad-CAM++ heatmaps from the final convolutional layer.

Figure 9: Raw RGB, XYZ input frames and Grad-CAM++ activations on the Open Drawer RoboCasa task for RGB-only and Point-map-only ConvNeXtv2 visual encoders.

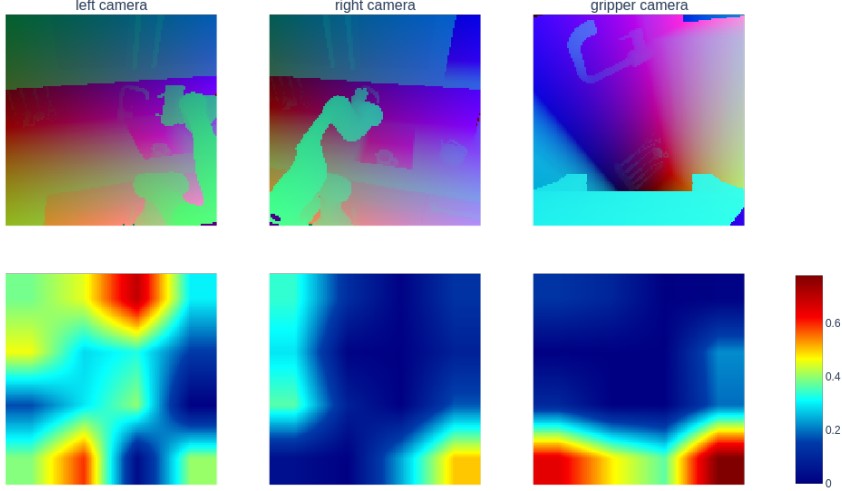

(a) RGB-only ConvNeXtv2 encoder. Top: raw 128x128 RGB input frames provided to the agent. Bottom: Grad-CAM++ heatmaps from the final convolutional layer.

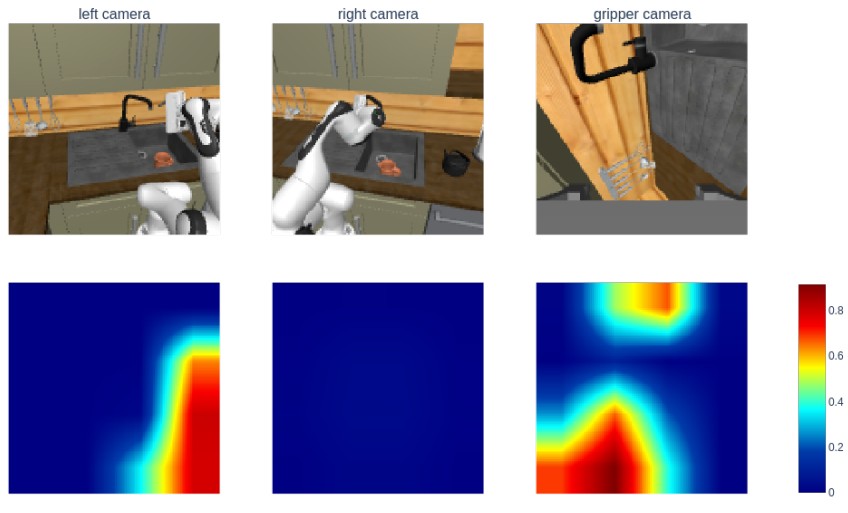

(b) PMP-xyz ConvNeXtv2 encoder. Top: XYZ input visualized as RGB. Bottom: Grad-CAM++ heatmaps from the final convolutional layer.

Figure 10: Raw RGB, XYZ input frames and Grad-CAM++ activations on the Turn On Sink Faucet RoboCasa task for RGB-only and Point-map-only ConvNeXtv2 visual encoders.

# E Compute Resources

For the CALVIN experiments, PMP-Cat employs Film-ResNet50 as encoders for both RGB images and point maps, with 10 x-Blocks as backbones (512 latent dimensions), totaling 147M trainable parameters. Training utilizes 4 Nvidia RTX 6000 Ada GPUs with 128 samples per GPU (512 total batch size). Each epoch completes in approximately 13 minutes, allowing full training (25 epochs) in under 6 hours, excluding evaluation time.

For the RoboCasa experiments, PMP-Cat employs ConvNeXtv2 as encoders with 8 x-Blocks using 512 latent dimensions. Training utilizes 1 NVIDIA A100-SXM4-40GB with a 128 batch size.

For the real-robot experiments, PMP-Cat employs Film-ResNet50 as encoders for both images and point maps, with 6 x-Blocks using 256 latent dimensions. Training utilizes 1 Nvidia RTX 6000 Ada GPUs with 128 batch size.

