# OpenReview forum: "PointMapPolicy: Structured Point Cloud Processing for Multi-Modal Imitation Learning"
_NeurIPS.cc/2025/Conference — NeurIPS 2025 poster_

### Official Review · Reviewer_QAhT · 2025-06-21

**Clarity:** 2
**Significance:** 2
**Originality:** 3
**Rating:** 4
**Confidence:** 4

**Summary:**

This paper proposes a multi-modal imitation learning method, namely PointMapPolicy. It uses point maps to organize raw point cloud data. The diffusion policy for action generation is integrated based on the EDM framework. The authors have tested their proposed method in both simulation and real-world environments. The main results demonstrate that the proposed PointMapPolicy outperforms the current 2D and 3D imitation learning baselines. The ablation studies also present the effectiveness of some devised components in the point-map-based scheme.

**Questions:**

The authors should address the concerns listed in the Weaknesses section. Here are some suggestions:

* The authors should clarify their motivation more clearly by providing the weaknesses of the current schemes in the introduction.

* Please consider modifying the existing diffusion-based imitation learning scheme to better incorporate point maps, rather than using the off-the-shelf ones.

* The definition of the point map should be provided.

* Please provide key ablation studies as mentioned in Major Weaknesses.

* When the authors propose using some configurations such as the number of Transformer layers, please clarify the motivation or provide related ablations.

**Ethical Concerns:**

["NO or VERY MINOR ethics concerns only"]

**Final Justification:**

The authors have addressed all my concerns. I will raise my rating to borderline accept.

**Limitations:**

yes

**Quality:**

2

**Strengths And Weaknesses:**

Strengths:

* This paper is easy to follow.
* The citations are proper, and the related works are well reviewed.
* Using point maps to organize point clouds for imitation learning is interesting.
* The proposed method is compared with multiple baseline methods in both simulation and real-world setups, providing extensive experimental results.

Weaknesses:

Major:

* While the introduction provides an overview of existing schemes, it does not clearly include their limitations, particularly the down-sampling-based scheme and feature-lifting scheme. This obscures the gaps that the proposed point-map-based method aims to address in the context. In addition, Figure 1, as one of the most important figures in the paper, is not referenced in the main text.

* The contributions of this work are limited. The only major contribution is using point maps as the input for imitation learning. The action diffusion module is off-the-shelf.

* The current formulation of Equation (3) provides only how to get point maps. However, a rigorous mathematical definition of the point map should be explicitly specified, since it is the major contribution of this work.

* In Table 2, PMP-xyz shows very poor performance compared to other baselines. Does this demonstrate that the whole PMP approach actually works mostly because of RGB input? The reviewer thinks that point maps struggle to capture useful geometric patterns since only using xyz cannot lead to good performance.

* Key ablation studies are missing. The authors should replace the point map encoding with the current point-cloud-format encoding (e.g., down-sampling-based encoding) to demonstrate its superiority.

* There is no analysis on some experimental results. For example, in Fusion of Images and Point Maps, the reason why Cat consistently emerges as the most effective fusion approach should be provided.

Minor:

* How to determine the number of denoising steps in PMP?
* Why have the authors used Film-ResNet rather than CLIP or other backbones to encode RGB images?
* Figure 3 omits the illustration of the concatenation operation.
* Why do the authors use a 'four-layer' transformer for the Attn scheme?

---

> ### Author Rebuttal · Authors · 2025-07-30
>
> We thank the reviewer for taking the time to review our work and for the many helpful comments and suggestions. We hope the following replies address the questions and concerns raised.
>
> ---
>
> > While the introduction provides an overview of existing schemes, it does not clearly include their limitations, particularly the down-sampling-based scheme and feature-lifting scheme. This obscures the gaps that the proposed point-map-based method aims to address in the context. In addition, Figure 1, as one of the most important figures in the paper, is not referenced in the main text. …… The authors should clarify their motivation more clearly by providing the weaknesses of the current schemes in the introduction.
> >
>
> We thank the reviewer for the important feedback and apologize for the missing reference to Figure 1. Here we provide a clear motivation and will add it in the final version:
>
> Current 3D processing approaches face fundamental limitations that create a critical gap between 3D geometric information and existing 2D vision architectures (Figure 1). **Downsampling-based methods** (Figure 1a) [1] suffer from an inherent information-fidelity tradeoff: they must dramatically reduce point density through techniques like Furthest Point Sampling to remain computationally tractable, inevitably discarding fine-grained geometric details essential for precise manipulation tasks. **Feature-lifting approaches** (Figure 1b) [2] face equally problematic limitations as they aggregate 2D features through depth averaging and 3D transformations, introducing information loss while struggling to maintain spatial structure during the lifting process.
>
> Our point map approach (Figure 1c) directly addresses both limitations by preserving complete spatial information in a structured 2D format, enabling direct application of established vision architectures without sacrificing geometric fidelity or computational efficiency. This representation maintains natural correspondence between visual appearance and geometric structure, eliminating the traditional tradeoff between precision and tractability.
>
> ---
>
> > The contributions of this work are limited. The only major contribution is using point maps as the input for imitation learning. The action diffusion module is off-the-shelf. …… Please consider modifying the existing diffusion-based imitation learning scheme to better incorporate point maps, rather than using the off-the-shelf ones.
> >
>
> We thank the reviewer for this concern. We clarify that our work is the first to systematically validate point maps for end-to-end imitation learning using diffusion policies. Our contribution is introducing point maps as a representation that enables any policy framework to leverage 3D geometric information through standard vision encoders.
> Just as RGB images work with any vision architecture (ResNet, ViT, ConvNeXt), point maps can be directly processed by existing encoders without modification. This eliminates the need for specialized point cloud networks and enables seamless integration into any visual imitation learning framework. Our systematic validation demonstrates the effectiveness of this representation across diverse tasks and benchmarks, providing evidence that enables the broader community to adopt point maps in their preferred policy architectures.
>
> ---
>
> > The current formulation of Equation (3) provides only how to get point maps. However, a rigorous mathematical definition of the point map should be explicitly specified, since it is the major contribution of this work.
> >
>
> We really appreciate this important suggestion. We provide the definition of the Point Map and will add it to the final version of the paper:
>
> A Point Map $X \in \mathbb{R}^{W \times H \times 3}$ is a dense 2D field of 3D points that establishes a one-to-one mapping between image pixels and 3D scene points. For an RGB image $I$ of resolution $W \times H$, the corresponding Point Map $X$ satisfies $I_{i,j} \leftrightarrow X_{i,j}$ for all pixel coordinates $(i,j) \in \\{1 \ldots W\\} \times \\{1 \ldots H\\}$, where each pixel intensity $I_{i,j}$ corresponds to a 3D point $X_{i,j} \in \mathbb{R}^3$ in world coordinates. This representation assumes that each camera ray intersects a single 3D point, ignoring the case of translucent surfaces [3].
>
> ---
>
> > In Table 2, PMP-xyz shows very poor performance compared to other baselines. Does this demonstrate that the whole PMP approach actually works mostly because of RGB input? The reviewer thinks that point maps struggle to capture useful geometric patterns since only using xyz cannot lead to good performance.
> >
>
> We thank the reviewer for this important question. The performance gap between RGB and XYZ inputs on the CALVIN tasks is a task-dependent limitation, and not a limitation of point maps in general. As we discuss in lines 216-220, PMP-xyz's lower performance on CALVIN (2.03 vs RGB's 3.15) stems from CALVIN's heavy reliance on color information, with many instructions explicitly referencing colors (e.g., "red block," "pink block"). Without color cues, the model cannot complete color-dependent tasks regardless of geometric precision.
>
> However, this task-dependency actually validates our approach: different modalities excel at different tasks. On geometry-heavy RoboCasa tasks, PMP-xyz (49.12%) significantly outperforms RGB (40.16%), while on color-dependent CALVIN tasks, RGB dominates. Most importantly, our multi-modal PMP achieves the best of both worlds - outperforming RGB on both benchmarks (CALVIN: 4.01 vs 3.15; RoboCasa: 47.22% vs 40.16%) by adaptively leveraging the most relevant information for each task.
>
> This demonstrates the core value of our approach: complementary modalities that together provide more robust performance across diverse manipulation scenarios.
>
> ---
>
> > Key ablation studies are missing. The authors should replace the point map encoding with the current point-cloud-format encoding (e.g., down-sampling-based encoding) to demonstrate its superiority. …… Please provide key ablation studies as mentioned in Major Weaknesses.
> >
>
> We appreciate the reviewer’s valuable suggestions. In fact, Reviewer Dr9P raised similar concerns about separating input modality effects from policy backbone influences. We conducted extensive controlled ablations comparing different point cloud processing methods using identical xLSTM backbones. These results demonstrate that our improvements stem from the point map representation itself, not architectural choices. **We kindly refer the reviewer to our detailed response to Reviewer Dr9P for the complete ablation study.**
>
> ---
>
> > There is no analysis on some experimental results. For example, in Fusion of Images and Point Maps, the reason why Cat consistently emerges as the most effective fusion approach should be provided.
> >
>
> Cat fusion performs well because it preserves all information from both modalities without loss by directly concatenating all tokens as input to the diffusion policy, avoiding any intermediate processing that could discard useful features.
>
> ---
>
> > How to determine the number of denoising steps in PMP?
> >
>
> We follow the established practice from diffusion policy literature [4] and use 4 denoising steps, which provides a good balance between generation quality and computational efficiency.
>
> ---
>
> > Why have the authors used Film-ResNet rather than CLIP or other backbones to encode RGB images?
> >
>
> FiLM-ResNet is a standard choice for robotic vision tasks due to its effective conditioning mechanisms and has been established in prior work including RT-1 [5], MT-ACT [6], and MoDE [7].
>
> ---
>
> > Figure 3 omits the illustration of the concatenation operation.
> >
>
> We will add concatenation illustration for clarity in the final version.
>
> ---
>
> > Why do the authors use a 'four-layer' transformer for the Attn scheme? …… When the authors propose using some configurations such as the number of Transformer layers, please clarify the motivation or provide related ablations.
> >
>
> We additionally provide ablation results on different transformer layer counts (4, 6, 8) for the Attn fusion scheme across three RoboCasa task categories. The results below show that increasing the number of attention layers does not improve performance.
>
> |  | Drawers | Buttons | Insertion |
> | --- | --- | --- | --- |
> | Attn-4 | $72.7$ | $75.3$ | $34.0$ |
> | Attn-6 | $69.6$ | $72.9$ | $29.0$ |
> | Attn-8 | $68.0$ | $72.0$ | $30.0$ |
>
> ---
>
> We sincerely thank the reviewer for providing specific and constructive suggestions to strengthen our work, and hope that our responses have successfully addressed the questions and concerns raised. Should the reviewer have any additional questions or further concerns, we would be more than happy to provide further clarifications or answers.
>
> ---
>
> References
>
> [1] Ze, Yanjie, et al. "3d diffusion policy: Generalizable visuomotor policy learning via simple 3d representations." RSS 2024
>
> [2] Ke, Tsung-Wei, Nikolaos Gkanatsios, and Katerina Fragkiadaki. "3d diffuser actor: Policy diffusion with 3d scene representations." CoRL 2024
>
> [3] Wang, Shuzhe, et al. "Dust3r: Geometric 3d vision made easy." CVPR 2024
>
> [4] Reuss, Moritz, et al. "Goal-conditioned imitation learning using score-based diffusion policies." RSS 2023.
>
> [5] Brohan, Anthony, et al. "Rt-1: Robotics transformer for real-world control at scale." RSS 2023.
>
> [6] Bharadhwaj, Homanga, et al. "Roboagent: Generalization and efficiency in robot manipulation via semantic augmentations and action chunking." ICRA 2024.
>
> [7] Reuss, Moritz, et al. "Efficient diffusion transformer policies with mixture of expert denoisers for multitask learning." ICLR 2025

---

> > ### Author Response · Authors · 2025-08-05
> >
> > Dear Reviewer QAhT,
> >
> > As the discussion period is ending soon, we would greatly appreciate it if you could let us know whether our response addressed all of your remaining concerns. If you have any remaining questions or points that require clarification, we are happy to provide additional information. Thanks a lot for your suggestions for improving our paper!

---

> > > ### Comment · Reviewer_QAhT · 2025-08-08
> > >
> > > Thanks for your comprehensive clarifications. However, I still have some concerns:
> > >
> > > (1) The rebuttal fails to address the core concern about whether point maps genuinely improve geometric reasoning. The performance gap between PMP-xyz and RGB inputs suggests that the method's success may heavily rely on color information, undermining its claimed geometric benefits. Although the authors attribute this to "task-dependent limitation", it is not convincing enough, as the point-map-based method should be robust enough to address the task-dependent limitation.
> > >
> > > (2) The contributions of this work are limited. I do not think only using point maps as the input for imitation learning without adapting other modules to this modality can be regarded as a convincing major contribution.
> > >
> > > (3) The analysis of fusion strategies lacks depth, failing to explain why concatenation outperforms attention-based fusion.

---

> ### Author Response · Authors · 2025-08-08
>
> Dear Reviewer QAhT,
>
> Thank you very much for your detailed review. We appreciate your continued engagement and would like to address the remaining concerns:
>
> (1) Regarding geometric reasoning capabilities, many manipulation tasks, particularly in CALVIN, explicitly require color information to identify target objects (e.g., "pick up the red block", "move the pink cube"). For such tasks, it is fundamentally impossible for any purely geometric representation (xyz) to succeed, regardless of its sophistication. This explains PMP-xyz's lower performance on CALVIN, but importantly, this is not a limitation of the point map representation itself.
>
> This is further validated by examining scenarios where geometric information is sufficient: on RoboCasa tasks, which primarily require spatial understanding, **PMP-xyz (49.12%) significantly outperforms RGB-only approaches (40.16%)**. This demonstrates that point maps effectively capture and leverage geometric information when color cues are not critical. The key strength of our approach lies in combining both modalities. While xyz alone cannot solve color-dependent tasks (which we never claimed it should), it provides crucial complementary geometric information that enhances overall performance when combined with RGB. This is evidenced by **our full PMP method consistently outperforming RGB-only baselines across both benchmarks - improving performance on CALVIN (4.01 vs 3.15) and RoboCasa (47.22% vs 40.16%)**. These results demonstrate that point maps successfully enrich policies with valuable geometric information while preserving the benefits of RGB input.
>
> (2) Our contribution extends beyond simply using point maps as input:
>
> - systematic validation demonstrating point maps' effectiveness for imitation learning across diverse tasks and modalities
> - rigorous experimental methodology comparing point maps with other 3D representations using identical architectures to isolate representational benefits
> - demonstration that standard vision encoders can directly process point maps without architectural modifications
> - practical insights on fusion strategies enabling broader adoption of this representation in robotic learning systems.
>
> (3) Thank you for raising this point about the fusion analysis. We acknowledge that our use of the term "concatenation" was potentially misleading. What we labeled as "Cat" fusion actually means directly feeding both point map and RGB tokens (6 tokens for 3 cameras in total) into the xLSTM-based policy, allowing the xLSTM itself to perform the fusion. This implicit fusion through xLSTM performs better than explicit transformer-based attention fusion, which is an important insight about the effectiveness of letting the policy network handle the multimodal fusion directly. We will rename this fusion method from "Cat" to "xLSTM fusion" in the final version of the paper to better reflect its actual mechanism and avoid confusion.
>
> We hope this addresses the reviewer’s remaining concerns and would be happy to provide further clarifications or additional information if needed.

---

> > ### Comment · Reviewer_QAhT · 2025-08-08
> >
> > Thanks for addressing all my concerns. I will raise my rating.

---

> > > ### Author Response · Authors · 2025-08-09
> > >
> > > Dear Reviewer QAhT,
> > >
> > > Thank you very much for your thoughtful engagement throughout this process and for taking the time to consider our responses. We sincerely appreciate your willingness to raise your rating and are grateful for the constructive feedback that has helped improve our work.

---

### Official Review · Reviewer_r4FG · 2025-07-01

**Clarity:** 3
**Significance:** 3
**Originality:** 3
**Rating:** 5
**Confidence:** 3

**Summary:**

This paper attempts to augment vision-based policies with 3D information to improve the motion accuracy and hence the task success rate. Unlike previous works exploring point clouds or depth-image, the paper suggests using point map, converted from depth image via known camera intrinsic parameters. The main benefit is the possibility of using a similar image encoder and position embeddings as the RGB modality, in contrast to point cloud data. The approach explores different ways of fusing point map and RGB encodings, identifying the benefits of concatenation. The design is utilised in training a language prompted diffusion policy, leveraging xLSTM for light-weight encoding and training of the history. The approach is compared to several baselines with/without input carrying 3D information on two simulation benchmarks and real robot experiments. The results show improvement on the task success rate and the length of following the language instructions.

**Questions:**

* Can more insights be shed on why point map brings more performance gains comparing to other 3D especially depth map? It seems like depth map information is just distorted with a scale by not knowing the camera intrinsics. Any other factors may contribute to the performance gap?
* What is the implication for the extra requirement of knowing camera intrinsics? Would that be a limitation for utilising large-scale video action data without camera parameters?
* Can additional ablation study be done to show the significance of the xLSTM design choice?

**Ethical Concerns:**

["NO or VERY MINOR ethics concerns only"]

**Final Justification:**

The authors' additional ablation helps to clarify some confusions in the remark. The significance of the proposed PointMap seems eclipsed somehow by the xLSTM used in peer works. But overall the paper provides an idea that seems simple yet effective for learning 3D vision-based manipulation.

**Limitations:**

Yes.

**Paper Formatting Concerns:**

No.

**Quality:**

3

**Strengths And Weaknesses:**

Strengths:
* Simple yet effective design to fuse 3D information in vision-policy training.
* The motivation for having a unified visual encoding architecture for capturing 3D information is reasonable.
* Extensive studies with multiple baselines and benchmarks, including real-robot experiments.
* The design has the potential to augment multiple views/sensors input, for the ease of manipulating image encodings.

Weakness:
* Though the empirical performance is positive to the main hypothesis, more insightful discussion or studies on why point map is favoured, especially comparing to depth image, is lacking.
* The significance of xLSTM is not very clear in the comparison.

---

> ### Author Rebuttal · Authors · 2025-07-30
>
> We thank the reviewer for taking the time to review our work and for the many helpful comments and suggestions. We hope the following replies address the questions and concerns raised.
>
> ---
>
> > Though the empirical performance is positive to the main hypothesis, more insightful discussion or studies on why point map is favoured, especially comparing to depth image, is lacking.
> Can more insights be shed on why point map brings more performance gains comparing to other 3D especially depth map? It seems like depth map information is just distorted with a scale by not knowing the camera intrinsics. Any other factors may contribute to the performance gap?
> >
>
> We thank the reviewer for this question. Point maps provide explicit 3D world coordinates (X,Y,Z) while depth maps only provide distance from camera. This difference is crucial for manipulation tasks. Point maps enable the network to directly reason about object positions, orientations, and spatial relationships in world coordinates, while depth requires the network to implicitly learn the camera projection model. Beyond camera Intrinsics, the transformation also handles extrinsic parameters (camera pose), enabling fusion of multi-view point maps in a common reference frame - something not directly possible with depth maps.
>
> ---
>
> > What is the implication for the extra requirement of knowing camera intrinsics? Would that be a limitation for utilising large-scale video action data without camera parameters?
> >
>
> We thank the reviewer for raising this important practical consideration. Indeed, the camera parameter requirement limits applicability to large-scale uncalibrated datasets. However, point maps serve as an alternative to other 3D representations, which also require camera calibration, so this limitation is not unique to our approach. Additionally, recent foundational models like VGGT [1] can infer 3D attributes, including camera parameters and point maps, from just a few views. By leveraging such methods, we could generate point maps from any 2D dataset, opening possibilities for large-scale pretraining of point map encoders. We will add this promising direction to our future work discussion in the final version.
>
> ---
>
> > The significance of xLSTM is not very clear in the comparison. …… Can additional ablation study be done to show the significance of the xLSTM design choice?
> >
>
> Our xLSTM choice is motivated by X-IL [2], which systematically compared xLSTM vs Transformers in multi-modal imitation learning, showing xLSTM's advantages for both RGB and point cloud inputs. However, we agree that additional ablation would strengthen our work since their comparison didn't include point map representations. We conducted controlled experiments comparing xLSTM vs Transformer-based PointMapPolicy using matched model sizes and identical training schemes on a subset of RoboCasa tasks:
>
> |  | PMP-xyz (Transformer) | PMP-xyz (xLSTM) |
> | --- | --- | --- |
> | TurnOnSinkFaucet | $50.7 \scriptstyle{\pm 8.1}$ | $\mathbf{76.7 \scriptstyle{\pm 4.1}}$ |
> | TurnOffSinkFaucet | $52.7 \scriptstyle{\pm 10.5}$ | $\mathbf{82.0 \scriptstyle{\pm 1.6}}$ |
> | TurnSinkSpout | $45.3 \scriptstyle{\pm 10.9}$ | $\mathbf{76.0 \scriptstyle{\pm 1.6}}$ |
> | CoffeePressButton | $80.0 \scriptstyle{\pm 2.8}$ | $\mathbf{82.7 \scriptstyle{\pm 5.0}}$ |
> | TurnOnMicrowave | $41.3 \scriptstyle{\pm 11.5}$ | $\mathbf{49.3 \scriptstyle{\pm 5.0}}$ |
> | TurnOffMicrowave | $50.0 \scriptstyle{\pm 10.7}$ | $\mathbf{70.0 \scriptstyle{\pm 4.9}}$ |
> | CoffeeServeMug | $56.0 \scriptstyle{\pm 5.9}$ | $\mathbf{69.3 \scriptstyle{\pm 6.6}}$ |
> | CoffeeSetupMug | $14.7 \scriptstyle{\pm 3.4}$ | $\mathbf{16.7 \scriptstyle{\pm 3.8}}$ |
>
> The results demonstrate that xLSTM consistently outperforms Transformer-based policies when using point map representations, validating our architectural choice and suggesting that xLSTM's efficiency advantages extend to our structured 3D representation.
>
> ---
>
> We would like to thank the reviewer again and welcome the opportunity to address any additional concerns or questions that the reviewer may have.
>
> ---
>
> References
>
> [1] Wang, Jianyuan, et al. "Vggt: Visual geometry grounded transformer." *Proceedings of the Computer Vision and Pattern Recognition Conference*. 2025.
>
> [2] Jia, Xiaogang, et al. "X-il: Exploring the design space of imitation learning policies." arXiv preprint arXiv:2502.12330 (2025).

---

> > ### Comment · Reviewer_r4FG · 2025-08-01
> >
> > Thanks for the responses and extra ablations. The improvement caused by xLSTM looks significant in certain tasks. How will this explain away the benefits brought by the introduction of PointMap? It seems the performance on SinkFaucet is even lower than using RGB/Depth modalities? Does this mean it is actually xLSTM doing the heavy lifting here?

---

> > > ### Author Response · Authors · 2025-08-03
> > >
> > > Thank you for this insightful follow-up question. We want to first clarify that in our main Table 1 results, all modalities (RGB, Depth, PMP-xyz, PMP) use identical xLSTM architectures - so the point map advantages there are purely from representation, not architecture differences.
> > >
> > > The xLSTM vs Transformer comparison shows architecture effects, but this doesn't diminish the point map contribution - rather, it demonstrates that both representation and architecture choices matter. We conducted additional experiments comparing Transformer across modalities on a subset of RoboCasa tasks:
> > >
> > > |  | RGB (Transformer) | Depth (Transformer) | PMP-xyz (Transformer) |
> > > | --- | --- | --- | --- |
> > > | TurnOnSinkFaucet | $60.0 \scriptstyle{\pm 4.3}$ | $61.7 \scriptstyle{\pm 5.7}$ | $50.7 \scriptstyle{\pm 8.1}$ |
> > > | TurnOffSinkFaucet | $52.7 \scriptstyle{\pm 7.4}$ | $56.7 \scriptstyle{\pm 7.7}$ | $52.7 \scriptstyle{\pm 10.5}$ |
> > > | TurnSinkSpout | $13.3 \scriptstyle{\pm 8.2}$ | $55.3 \scriptstyle{\pm 1.9}$ | $45.3 \scriptstyle{\pm 10.9}$ |
> > > | CoffeePressButton | $56.0 \scriptstyle{\pm 6.5}$ | $85.3 \scriptstyle{\pm 3.8}$ | $80.0 \scriptstyle{\pm 2.8}$ |
> > > | TurnOnMicrowave | $26.7 \scriptstyle{\pm 6.6}$ | $38.0 \scriptstyle{\pm 4.3}$ | $41.3 \scriptstyle{\pm 11.5}$ |
> > > | TurnOffMicrowave | $68.0 \scriptstyle{\pm 5.9}$ | $55.3 \scriptstyle{\pm 4.7}$ | $50.0 \scriptstyle{\pm 10.7}$ |
> > > | CoffeeServeMug | $32.7 \scriptstyle{\pm 6.2}$ | $46.0 \scriptstyle{\pm 4.3}$ | $56.0 \scriptstyle{\pm 5.9}$ |
> > > | CoffeeSetupMug | $16.7 \scriptstyle{\pm 2.5}$ | $14.0 \scriptstyle{\pm 1.6}$ | $14.7 \scriptstyle{\pm 3.4}$ |
> > > | Average | $40.76$ | $51.53$ | $48.83$ |
> > >
> > > Since these tasks are among the easier RoboCasa tasks, requiring less 3D geometric reasoning capability, depth, and point maps perform similarly on these tasks. Note that depth and point maps are actually within error of each other for each of these tasks. Even RGB (without depth) performs well on some of these tasks, suggesting that they may be solvable without geometric reasoning at all. We were not yet able to test Transformer backbones for the more difficult tasks, such as PnPCounterToMicrowave, PnPMicrowaveToCounter, PnPSinkToCounter, TurnOnStove, and CoffeeServeMug, but we will include these in the camera-ready version. When comparing depth and point maps on these tasks with the xLSTM backbone (Table 1 in the manuscript), point maps consistently outperform depth. This suggests that point maps provide more reliable performance as task complexity increases.
> > >
> > > Furthermore, we provided extended experiments during the rebuttal comparing point maps against other point cloud processing methods (PointNet-xyz, PointNet-color, PointPatch, 3D-Lifting) using identical xLSTM backbones. When the backbone is kept constant, point maps consistently outperform all alternatives, further confirming representation advantages. **We kindly refer the reviewer to our detailed response to Reviewer Dr9P for the complete ablation study.**
> > >
> > > We hope this addresses the reviewer’s remaining concerns and would be happy to provide further clarifications or additional information if needed.

---

### Official Review · Reviewer_Dr9P · 2025-07-01

**Clarity:** 2
**Significance:** 3
**Originality:** 1
**Rating:** 4
**Confidence:** 5

**Summary:**

This paper proposes PointMapPolicy for diffusion-based imitation learning in robotic manipulation. Its core contribution is to use a Point Map observation that projects the XYZ coordinates of a point cloud back onto the image plane, supplying rich geometric cues for policy learning. Extensive experiments in both simulation and the real world show that PointMapPolicy outperforms existing baselines.

**Questions:**

1. As mentioned in Weakness 2, separating the evaluation of the input modality and the policy backbone would strengthen the paper. Although Table 1 shows that the proposed method is outperforming DP3 and 3DA, is it because Point Map is a better input modality, or because the X-Block is better than the UNet used in DP3 and the transformer used in 3DA? Adding an additional baseline that uses a point cloud encoder + X-Block variation would be beneficial.
2. Moreover, another baseline that uses 3D Lifting + X-Block will also better demonstrate the advantages of Point Map, particularly because 3D Lifting is listed as a comparison in Figure 1.
3. A clearer contribution summary would be useful. How is the proposed method different from other robot learning approaches that use a similar Point Map concept, like RVT?
4. Why do the authors list their model as no pre-training in Table 2? Figure 2 states `PMP integrates multiple modalities: language instructions encoded by a pretrained CLIP model, images processed by pretrained visual encoders, and point maps processed by visual encoders trained from scratch.`. Line 198 seems to argue that the proposed method does not use `robot-specific pretraining`, but it is important to clarify what is the difference here, especially because the proposed method underperforms the pretrained baselines in Table 2.
5. The overall writing could be polished; for instance, the abstract does not read very smoothly.

**Ethical Concerns:**

["NO or VERY MINOR ethics concerns only"]

**Final Justification:**

Most of my concerns are addressed by the authors. The new experiment clearly demonstrates the advantages of using point maps as the input modality for policy learning. While I have reservations about the novelty of the paper, I partially agree with the authors’ claims and arguments, so I increase my rating to borderline accept.

**Limitations:**

Yes

**Paper Formatting Concerns:**

Should the table caption be moved to the top of the table?

**Quality:**

2

**Strengths And Weaknesses:**

Strength
1. The paper tackles input-modality selection in robotic manipulation, a very important problem.
2. The proposed point-map representation can be easily adopted by different policy-learning frameworks.
3. I appreciate the experimental videos on the website, especially those testing diverse initial configurations.

Weakness
1. The technical contribution appears incremental. Point Maps are an existing concept [10, 11], and prior work has used similar inputs in manipulation. E.g., RVT uses a similar input format: ```Specifically, for each view, we render three image maps with a total of 7 channels: (1) RGB (3 channels), (2) depth (1 channel), and (3) (x,y,z) coordinates of the points in the world frame (3 channels).```
2. The goal is to advocate Point Maps as an input modality, but the experimental justification is not rigorous. Specifically, Table 1 mixes input modality with backbone architecture, where PMP uses an X-Block, while DP3 (UNet) and 3DA (Transformer) use different policy backbone, making it unclear whether the improvement come from the modality or the policy backbone.
3. Some alternative point-cloud approaches, such as the 3D-Lifting method in Figure 1, are also missing.

---

> ### Author Rebuttal · Authors · 2025-07-30
>
> We thank the reviewer for taking the time to review our work and for the many helpful comments and suggestions. We hope the following replies address the questions and concerns raised.
>
> ---
>
> > The technical contribution appears incremental. Point Maps are an existing concept [10, 11], and prior work has used similar inputs in manipulation. E.g., RVT uses a similar input format
> ……. A clearer contribution summary would be useful. How is the proposed method different from other robot learning approaches that use a similar Point Map concept, like RVT?
> >
>
> We thank the reviewer for this important concern. While Point Maps exist as a concept in 3D vision, our contribution lies in demonstrating their effectiveness for general imitation learning. We provide a clear summary of our contributions and key differences from RVT:
>
> - RVT is a learning-based motion planner that predicts keyframe poses and relies on traditional motion planners for execution. In contrast, PMP is the first to apply point maps to end-to-end behavior cloning using diffusion policies, enabling direct action prediction without separate planning stages.
> - While RVT uses point map-like inputs, it makes a number of other design decisions that make it hard to disentangle the benefits of point maps compared to other 3D representations. For instance, they re-render a point cloud with virtual cameras, ignoring self-occlusions by unobserved geometry. Our extensive controlled experiments across modalities with identical backbones provide the first rigorous evidence that point maps outperform traditional 3D representations in robotic learning contexts.
> - RVT naively concatenates geometric and color information at the channel level, while we systematically investigate multiple fusion strategies (Add/Cat/Attn) and demonstrate their robustness.
>
> ---
>
> > Some alternative point-cloud approaches, such as the 3D-Lifting method in Figure 1, are also missing.
> >
>
> We apologize for any confusion about our presentation regarding the 3D-Lifting methods. **3D Diffuser Actor (3DA) is indeed the 3D-Lifting method** shown in Figure 1(b), as stated in lines 70-72: "3D Diffuser Actor computes tokens by lifting 2D image features into 3D space".  Additionally, the Downsampling-based method in Figure 1(a) refers to DP3, as stated in lines 67-68: “DP3 encodes sparse point clouds using FPS”. We will make the baseline descriptions clearer in the final version.
>
> ---
>
> > Table 1 mixes input modality with backbone architecture, where PMP uses an X-Block, while DP3 (UNet) and 3DA (Transformer) use different policy backbone, making it unclear whether the improvement come from the modality or the policy backbone. …… separating the evaluation of the input modality and the policy backbone would strengthen the paper. …… Adding an additional baseline that uses a point cloud encoder + X-Block variation would be beneficial. …….. Moreover, another baseline that uses 3D Lifting + X-Block will also better demonstrate the advantages of Point Map
> >
>
> We thank the reviewer for this valuable suggestion and agree on the need for more controlled comparisons.
>
> **Additional Controlled Ablations:** To directly address backbone vs. modality effects, we conducted controlled experiments on RoboCasa using **identical xLSTM backbones** with different point cloud processing encoders: 1) **PointNet-xyz**: Following DP3, we gather point clouds from 3 camera views and use Furthest Point Sampling (FPS) to downsample to 1024 points, then apply MLP with maxpooling to create a compact 3D token. 2) **PointNet-color**: Same process as PointNet-xyz but using colored points with XYZRGB information. 3) **PointPatch**: We use FPS to sample 256 center points, apply k-Nearest Neighbors to create 256 point patches with 32 points each, tokenize each patch using MLP with maxpooling, then process the resulting tokens with a transformer to generate compact 3D representations. 4) **3D-Lifting**: We extract CLIP features (frozen) from each camera view and lift the 2D features into 3D space, then use a transformer to process the lifted tokens. The 3D tokens are then passed to the diffusion policy with an identical X-Block backbone.
>
> |  | PointNet-xyz | PointNet-color | PointPatch | 3D-Lifting | PMP-xyz (ours) |
> | --- | --- | --- | --- | --- | --- |
> | PnPCounterToMicrowave | $0.7 \scriptstyle{\pm 0.9}$ | $1.3 \scriptstyle{\pm 0.9}$ | $0.7 \scriptstyle{\pm 0.9}$ | $3.3 \scriptstyle{\pm 1.9}$ | $\mathbf{13.3 \scriptstyle{\pm 3.4}}$ |
> | PnPCounterToSink | $1.3 \scriptstyle{\pm 0.9}$ | $0.7 \scriptstyle{\pm 0.9}$ | $0.7 \scriptstyle{\pm 0.9}$ | $0.7 \scriptstyle{\pm 0.9}$ | $\mathbf{6.7 \scriptstyle{\pm 2.5}}$ |
> | PnPMicrowaveToCounter | $0$ | $0$ | $0$ | $4.0 \scriptstyle{\pm 1.6}$ | $\mathbf{16.0 \scriptstyle{\pm 1.6}}$ |
> | PnPSinkToCounter | $0$ | $0$ | $0$ | $3.3 \scriptstyle{\pm 1.9}$ | $\mathbf{8.0 \scriptstyle{\pm 1.6}}$ |
> | OpenDrawer | $16.7 \scriptstyle{\pm 2.5}$ | $10.7 \scriptstyle{\pm 0.9}$ | $25.3 \scriptstyle{\pm 2.5}$ | $26.7 \scriptstyle{\pm 2.5}$ | $\mathbf{60.0 \scriptstyle{\pm 4.3}}$ |
> | CloseDrawer | $92.0 \scriptstyle{\pm 1.6}$ | $92.0 \scriptstyle{\pm 1.6}$ | $93.3 \scriptstyle{\pm 0.9}$ | $64.7 \scriptstyle{\pm 9.6}$ | $\mathbf{96.0 \scriptstyle{\pm 1.6}}$ |
> | TurnOnStove | $42.0 \scriptstyle{\pm 0.0}$ | $31.3 \scriptstyle{\pm 3.4}$ | $38.0 \scriptstyle{\pm 4.9}$ | $28.7 \scriptstyle{\pm 5.2}$ | $43.3 \scriptstyle{\pm 5.7}$ |
> | TurnOffStove | $19.3 \scriptstyle{\pm 1.9}$ | $19.3 \scriptstyle{\pm 1.9}$ | $19.3 \scriptstyle{\pm 5.0}$ | $14.7 \scriptstyle{\pm 2.5}$ | $20.0 \scriptstyle{\pm 3.3}$ |
> | TurnOnSinkFaucet | $43.3 \scriptstyle{\pm 4.1}$ | $28.7 \scriptstyle{\pm 3.8}$ | $32.0 \scriptstyle{\pm 0.0}$ | $30.0 \scriptstyle{\pm 5.9}$ | $\mathbf{76.7 \scriptstyle{\pm 4.1}}$ |
> | TurnOffSinkFaucet | $50.0 \scriptstyle{\pm 7.1}$ | $51.3 \scriptstyle{\pm 3.4}$ | $48.7 \scriptstyle{\pm 5.7}$ | $41.3 \scriptstyle{\pm 8.4}$ | $\mathbf{82.0 \scriptstyle{\pm 1.6}}$ |
> | TurnSinkSpout | $62.0 \scriptstyle{\pm 1.6}$ | $64.0 \scriptstyle{\pm 1.6}$ | $58.7 \scriptstyle{\pm 3.4}$ | $64.0 \scriptstyle{\pm 0.0}$ | $\mathbf{76.0 \scriptstyle{\pm 1.6}}$ |
> | CoffeePressButton | $8.0 \scriptstyle{\pm 1.6}$ | $8.7 \scriptstyle{\pm 5.7}$ | $8.0 \scriptstyle{\pm 3.3}$ | $10.0 \scriptstyle{\pm 4.3}$ | $\mathbf{82.7 \scriptstyle{\pm 5.0}}$ |
> | TurnOnMicrowave | $36.0 \scriptstyle{\pm 3.3}$ | $26.7 \scriptstyle{\pm 5.0}$ | $35.3 \scriptstyle{\pm 9.6}$ | $25.3 \scriptstyle{\pm 2.5}$ | $\mathbf{49.3 \scriptstyle{\pm 5.0}}$ |
> | TurnOffMicrowave | $48.0 \scriptstyle{\pm 0.0}$ | $39.3 \scriptstyle{\pm 5.2}$ | $46.7 \scriptstyle{\pm 4.1}$ | $28.0 \scriptstyle{\pm 7.1}$ | $\mathbf{70.0 \scriptstyle{\pm 4.9}}$ |
> | CoffeeServeMug | $12.0 \scriptstyle{\pm 2.8}$ | $9.3 \scriptstyle{\pm 0.9}$ | $12.7 \scriptstyle{\pm 0.9}$ | $24.7 \scriptstyle{\pm 2.5}$ | $\mathbf{69.3 \scriptstyle{\pm 6.6}}$ |
> | CoffeeSetupMug | $2.0 \scriptstyle{\pm 1.6}$ | $2.7 \scriptstyle{\pm 0.9}$ | $2.7 \scriptstyle{\pm 0.9}$ | $4.7 \scriptstyle{\pm 2.5}$ | $\mathbf{16.7 \scriptstyle{\pm 3.8}}$ |
> | Average | $27.08$ | $24.12$ | $26.38$ | $23.38$ | $\mathbf{49.12}$ |
>
> All methods are using the same training scheme as PMP-xyz, and we report the results from 3 seeds. Results show **PMP-xyz consistently outperforms** these alternatives with identical architectures, demonstrating that improvements stem from our point map representation rather than backbone choice.
>
> ---
>
> > Why do the authors list their model as no pre-training in Table 2? Figure 2 states `PMP integrates multiple modalities: language instructions encoded by a pretrained CLIP model, images processed by pretrained visual encoders, and point maps processed by visual encoders trained from scratch.`. Line 198 seems to argue that the proposed method does not use `robot-specific pretraining`, but it is important to clarify what is the difference here, especially because the proposed method underperforms the pretrained baselines in Table 2.
> >
>
> We apologize for the confusion and the missing definition of the pre-training (PrT) in Table 2. PrT in our paper refers to robot-specific pertaining, for example, MoDE [1] is pre-trained on a diverse mix of multi-robot datasets curated from the OXE dataset Collaboration; Seer [2] is pre-trained on large-scale robotic datasets, such as DROID.
>
> Regarding our PMP model, we clearly state:
>
> - RGB encoders use ImageNet pretraining (standard vision pretraining)
> - Point map encoders trained from scratch (no existing pretraining for XYZ inputs)
> - No robot demonstration pretraining (unlike Seer, RoboFlamingo, etc.)
>
> Performance Context: Compared to the Seer-Large scratch version (3.83) and pretraining version (4.28), our method achieves competitive results (4.01) without robot-specific pretraining.
>
> Additionally, we also discussed the potential of pre-training point maps in the Limitation and Future Work.
>
> ---
>
> > The overall writing could be polished; for instance, the abstract does not read very smoothly
> >
>
> We will carefully revise the abstract and manuscript to improve clarity, flow, and overall readability in the final version.
>
> ---
>
> > Should the table caption be moved to the top of the table?
> >
>
> Thank you for pointing this out. We will correct this formatting in the final version.
>
> ---
>
> Once again, we sincerely thank the reviewer for their many insightful questions, which have greatly contributed to improving the quality of our work. We welcome any further concerns or questions they may have and would be glad to address them.
>
> ---
>
> References
>
> [1]  Reuss, Moritz, et al. "Efficient diffusion transformer policies with mixture of expert denoisers for multitask learning." ICLR 2025
>
> [2] Tian, Yang, et al. "Predictive inverse dynamics models are scalable learners for robotic manipulation." ICLR 2025

---

> > ### Comment · Reviewer_Dr9P · 2025-08-01
> >
> > Thank you for the rebuttal, most of my concerns are addressed, and I appreciate this very complete new experiment in such a short time.
> >
> > I still have reservations about the novelty of the paper. I appreciate the authors discussion about the difference compared with RVT, however, both methods are policy learning methods and the main difference is only the open loop (RVT) vs closed loop (this paper) action space.
> >
> > That said, I do think the paper contributes valuable insight to the community, and I will raise my score accordingly.

---

> > > ### Author Response · Authors · 2025-08-03
> > >
> > > We sincerely thank the reviewer for acknowledging our extensive experimental work and for raising your score. We appreciate your recognition that our work provides valuable insights to the community.
> > >
> > > Regarding the novelty concern, we respectfully note that the difference extends beyond open-loop vs. closed-loop action spaces. Our key contributions include: (1) systematic validation that point maps are effective for imitation learning across diverse tasks and modalities, (2) rigorous experimental methodology with controlled comparisons using identical architectures, (3) demonstrating that existing vision encoders can directly process point maps without modification, and (4) practical insights on fusion strategies that enable broader adoption of this representation.
> > >
> > > While RVT demonstrates one application of point map-like inputs, our work provides the empirical foundation that enables the community to confidently adopt this representation. We believe this systematic validation and the practical insights we provide constitute a valuable contribution to the field.
> > >
> > > We hope this addresses the reviewer’s remaining concerns and would be happy to provide further clarifications or additional information if needed.

---

### Official Review · Reviewer_twzC · 2025-07-03

**Clarity:** 3
**Significance:** 3
**Originality:** 2
**Rating:** 4
**Confidence:** 4

**Summary:**

This paper proposes PointMapPolicy (PMP), a multi-modal fusion framework for imitation learning that integrates both RGB images and point maps. The authors compare different multi-modal fusion methods, including early fusion, late fusion, and attention-based fusion. Simulation and real-world experiments show that the proposed method outperforms policies trained using only RGB or Depth modalities.

**Questions:**

How does the proposed point map method compare to the point cloud method? The paper only compares to point cloud-based methods that have different backbones.

**Ethical Concerns:**

["NO or VERY MINOR ethics concerns only"]

**Limitations:**

How to maintain performance when one modality is performing well and one is not? For example, the two modalities have large success rate differences for the TurnSinkSpout task in RoboCasa, resulting in PMP performing worse than using either modality. The authors should discuss these kinds of edge cases and describe them in the limitations section. Additionally, Figure 5 doesn’t show a lot of difference in the different fusion methods.

**Quality:**

3

**Strengths And Weaknesses:**

Strength:
1. The paper is clearly written and well-structured, making it easy to follow.
2. The paper studies an underexplored area in robot manipulation: policy learning from 3D input.
3. The paper provides a thorough comparison across RGB, depth map, and point map input modalities. The paper highlights the benefits of combining them.


Weakness:
1. The visual encoder and the backbone in the proposed model are trained from scratch, while prior work shows that pretrained vision encoders can significantly boost performance. The authors should clarify their decision not to use pretrained encoders or pretrained backbones and discuss how existing pretrained models could be adapted to handle additional modalities like point maps.
2. The ablation study compares different visual encoder choices. However, the performance of imitation learning also depends on the backbone choice. It is unclear how much the performance gain is from multi-modal fusion itself versus the choice of backbone. More controlled experiments are needed to isolate the contribution of fusion.
3. The introduction states that point clouds struggle to capture fine-grained detail, motivating the use of point maps. However, this claim would be more convincing with a quantitative comparison between point clouds and point maps, considering the availability of convolutional networks like KPConv that process unstructured point clouds effectively.
4. The related work could be strengthened by discussing more recent approaches in multi-modal policy learning. In particular, the authors should consider comparing with:
Zhu, Yichen, Zhicai Ou, Feifei Feng, and Jian Tang. "Any2Policy: Learning Visuomotor Policy with Any-Modality." Advances in Neural Information Processing Systems 37 (2024): 133518-133540.

---

> ### Author Rebuttal · Authors · 2025-07-30
>
> We thank the reviewer for taking the time to review our work and for the many helpful comments and suggestions. We hope the following replies address the questions and concerns raised.
>
> ---
>
> > The visual encoder and the backbone in the proposed model are trained from scratch, while prior work shows that pretrained vision encoders can significantly boost performance. The authors should clarify their decision not to use pretrained encoders or pretrained backbones and discuss how existing pretrained models could be adapted to handle additional modalities like point maps.
> >
>
> We thank the reviewer for bringing this concern to our attention and apologize for any confusion. We train point map encoders from scratch because no existing pretrained models are designed for XYZ coordinate inputs. Standard ImageNet-pretrained models expect RGB inputs with specific statistical properties that don't apply to geometric coordinates. Our RGB encoders do use ImageNet pretraining (FiLM-ResNet50), which is standard practice [1,2,3,4]. For point maps, we show that training from scratch with ConvNeXtv2 consistently outperforms other modalities (Table 1), suggesting our approach effectively learns geometric features. We agree that developing pretrained models for point map representations is valuable future work - since point maps can leverage any existing visual encoder architecture (ResNet, ViT, etc.), they could potentially be pretrained using visual objectives (detection, segmentation) or directly on robot learning datasets.
>
> ---
>
> > The ablation study compares different visual encoder choices. However, the performance of imitation learning also depends on the backbone choice. It is unclear how much the performance gain is from multi-modal fusion itself versus the choice of backbone. More controlled experiments are needed to isolate the contribution of fusion. …… How does the proposed point map method compare to the point cloud method? The paper only compares to point cloud-based methods that have different backbones.
> >
>
> We thank the reviewer for suggesting ablations to demonstrate point map effectiveness. We ensure fair evaluation by using identical xLSTM architectures across all modalities. Table 1 shows PMP-xyz (49.12%) vs RGB (40.16%) and Depth (44.00%) with identical backbones, demonstrating 9% and 5% improvements respectively from modality choice, not architecture differences.
>
> Additionally, Reviewer Dr9P raised similar concerns about separating input modality effects from policy backbone influences. We conducted extensive controlled ablations comparing different point cloud processing methods using identical xLSTM backbones. These results demonstrate that our improvements stem from the point map representation itself, not architectural choices. **We kindly refer the reviewer to our detailed response to Reviewer Dr9P for the complete ablation study.**
>
> ---
>
> > The introduction states that point clouds struggle to capture fine-grained detail, motivating the use of point maps. However, this claim would be more convincing with a quantitative comparison between point clouds and point maps, considering the availability of convolutional networks like KPConv that process unstructured point clouds effectively.
> >
>
> While convolutional architectures for point clouds such as KPConv exist, they empirically perform worse than other architectures. Point Transformer [5] and its recent variant Point Transformer v3 [6], both attention-based architectures, significantly outperform KPConv. We compare against 3DA, which is also attention-based. Similarly, Point-BERT [7], a point patch architecture, outperforms KPConv. We present an ablation against a Point Patch architecture in our response to reviewer Dr9P, where PMP outperforms it. Given that convolutional methods are consistently shown to be inferior in the point cloud classification and segmentation literature, we opt to compare against attention-based and aggregation-based architectures, which show more promise.
>
> ---
>
> > The related work could be strengthened by discussing more recent approaches in multi-modal policy learning. In particular, the authors should consider comparing with: Zhu, Yichen, Zhicai Ou, Feifei Feng, and Jian Tang. "Any2Policy: Learning Visuomotor Policy with Any-Modality." Advances in Neural Information Processing Systems 37 (2024): 133518-133540.
> >
>
> We appreciate this reference and will add it to our related work. Any2Policy focuses on cross-modal policy transfer, while our contribution is specifically the structured point map representation that enables direct application of vision architectures to 3D data. These are complementary approaches.
>
> ---
>
> > How to maintain performance when one modality is performing well and one is not? For example, the two modalities have large success rate differences for the TurnSinkSpout task in RoboCasa, resulting in PMP performing worse than using either modality. The authors should discuss these kinds of edge cases and describe them in the limitations section. Additionally, Figure 5 doesn’t show a lot of difference in the different fusion methods.
> >
>
> We thank the reviewer for this insightful observation. Table 1 reveals an important limitation: while PMP shows significant improvements on tasks like PnPSinkToCounter, TurnOffMicrowave, and CoffeePressButton, it performs similarly or worse than individual modalities on others (e.g., TurnSinkSpout). This suggests that simple concatenation fusion may not be optimal when one modality dramatically outperforms the other.
>
> For instance, in the Turning Levers category, PMP-xyz substantially outperforms RGB, yet PMP performs similarly to RGB alone. This indicates the fusion mechanism may be dominated by the weaker modality rather than leveraging the stronger one. This represents a key limitation of our current approach that we will add to our limitations section. This analysis points toward important future research on adaptive fusion mechanisms, such as learned modality weights or dynamic modality selection, that could better exploit complementary strengths rather than averaging performance.
> Despite this limitation, concatenation fusion shows consistent advantages across diverse task categories (Figure 5), and PMP demonstrates substantial improvements over individual modalities on CALVIN (4.01 vs RGB: 3.15, PMP-xyz: 2.03), demonstrating the effectiveness of this approach.
>
> ---
>
> We express our gratitude to the reviewer for their valuable comments and suggestions. We are pleased to address any additional questions or concerns that may arise.
>
> ---
>
> References
>
> [1] Chi, Cheng, et al. "Diffusion policy: Visuomotor policy learning via action diffusion." *The International Journal of Robotics Research* (2023).
>
> [2] Nasiriany, Soroush, et al. "Robocasa: Large-scale simulation of everyday tasks for generalist robots." RSS 2024
>
> [3] Reuss, Moritz, et al. "Efficient diffusion transformer policies with mixture of expert denoisers for multitask learning." ICLR 2025
>
> [4] Wen, Junjie, et al. "Dexvla: Vision-language model with plug-in diffusion expert for general robot control." *arXiv preprint arXiv:2502.05855* (2025).
>
> [5] Zhao, Hengshuang, et al. "Point transformer." ICCV 2021.
>
> [6] Wu, Xiaoyang, et al. "Point transformer v3: Simpler faster stronger." CVPR 2024.
>
> [7] Yu, Xumin, et al. "Point-bert: Pre-training 3d point cloud transformers with masked point modeling." CVPR 2022.

---

> > ### Comment · Reviewer_twzC · 2025-08-07
> >
> > Thank you for the responses. All my concerns have been addressed. I appreciate the additional controlled ablation study that separates the influence of model architecture and modality. The additional ablation study shows the benefit of multi-modal fusion. This paper brings value to the area of policy learning from 3D input.

---

> > > ### Author Response · Authors · 2025-08-07
> > >
> > > Thank you very much for your thoughtful review and positive feedback. We are pleased to know that all your concerns have been addressed through our rebuttal. We greatly appreciate your recognition of the additional controlled ablation study and the value our work brings to policy learning from 3D input. We will incorporate your suggestions to improve the clarity of our paper and include the additional experiments in the final version.

---

### Author Response · Authors · 2025-08-09
**Thank you for the insightful discussions and help to improve the paper!**

Dear Reviewers,

We sincerely thank all of the reviewers for the thoughtful feedback and engaging discussions throughout the rebuttal process. Your insights have been invaluable in helping us improve our work on PointMapPolicy. Going into the reviewer discussion, we would like to summarize the main improvements that resulted from your feedback during the rebuttal:

- **Enhanced Motivation and Key Contributions**: Clarified existing method limitations and articulated our core contributions: systematic validation of point maps' effectiveness across diverse tasks and modalities, rigorous comparison with other 3D representations using controlled architectures, demonstration that standard vision encoders can process point maps directly, and practical fusion insights enabling broader adoption in robotic learning.
- **Rigorous Experimental Validation**: Added extensive controlled ablations comparing point maps against traditional point cloud processing methods (PointNet-xyz, PointNet-color, PointPatch, 3D-Lifting) using identical xLSTM architectures, demonstrating consistent point map advantages and isolating representation effects from backbone choices.
- **Technical Clarity and Definitions**: Provided mathematical definitions of point maps, clarified baseline descriptions, and explained key differences from RVT.
- **Limitations and Future Work**: Acknowledged key limitations including camera calibration requirements for large-scale datasets, simple concatenation fusion not being optimal when modalities have very different performance, and point map encoders being trained from scratch. Identified promising future directions including developing point map pretraining objectives, adaptive fusion mechanisms, and leveraging foundational models like VGGT for generating point maps from uncalibrated data.
- **Task-Dependency Analysis**: Demonstrated that point maps excel on manipulation tasks requiring spatial reasoning while acknowledging color-dependency limitations, showing that multi-modal fusion leverages complementary strengths effectively.
- **Writing Quality and Presentation**: Improved abstract flow, enhanced introduction structure, corrected table formatting, and polished overall presentation for better clarity and readability.

These changes have significantly strengthened our paper and we are committed to incorporating all these improvements into the final version. Thank you again for your time and valuable feedback!

Best regards,

The Authors

---

### Decision · Program_Chairs · 2025-09-17

**Decision:**

Accept (poster)

**Comment:**

The paper introduces a framework for imitation learning that integrates both RGB and depth information through point maps. The authors argue that point maps offer the advantage of a grid-like structure, allowing them to be processed with standard image backbones. Experiments on RoboCasa and CALVIN demonstrate strong performance compared to recent baselines. Reviewers acknowledged the paper’s contribution to advancing 3D policy learning, while noting the lack of direct, “apples-to-apples” comparisons with point cloud-based methods. These concerns were addressed in the rebuttal and subsequent discussion.